**Article** https://doi.org/10.1038/s41467-025-57214-w

# The global distribution patterns of alien vertebrate richness in mountains

Adrián García-Rodríguez [1] ✉, Bernd Lenzner [1], Julián A. Velasco [2], Anna Schertler [1], Ali Omer[1], Hanno Seebens [3,4], César Capinha [5,6], Belinda Gallardo [7], Stefan Dullinger [8] & Franz Essl[1]

The diverse biotas of the world's mountains face a challenging future due to increasing threats like climate change, land-use change, and biological invasions, the last being particularly understudied in these regions. Here we compile occurrence records for 717 alien vertebrate species distributed in 2984 mountains worldwide. We analyze their distribution, biogeographic origin, presence in protected areas, and the drivers' explaining alien vertebrate richness in mountains. We find that the alien vertebrates most frequently recorded are birds (318 species) and mammals (161 species) reported in 2595 and 1518 mountains globally, respectively. The Palearctic, Nearctic, and Australasian realms are the most common recipients; the Nearctic, Indo-Malay, and Afrotropic realms are the most frequent donors. Almost 50% of the alien species studied also occur in protected areas. Proxies of anthropogenic impacts (e.g., higher road density or lower biodiversity intactness) and mountains' physical characteristics (e.g., elevation range and roughness) explain the distribution of alien vertebrates in mountains. Importantly, the magnitude of invasions in tropical mountains could be underestimated due to sampling bias towards the Northern Hemisphere and Australia. Our large-scale assessment reveals the advance of alien vertebrates in mountains worldwide and urges attention to minimize the impacts of biological invasions on the exceptional mountain biotas.

Orogenic processes have endowed the Earth's surface with mountainous landscapes in which a mosaic of environments converges[1]. Mountains cover only a small portion of the global land surface, but their physical and climatic heterogeneity supports a disproportionately high species diversity[2,3]. For instance, mountains harbor high levels of endemism and numerous biodiversity hotspots, highlighting their importance for life on Earth[3–5].

Mountains' rich biotas reflect the major evolutionary roles played by these regions, both as "cradles" where species arise at a faster pace[6,7] and "museums" where biodiversity has accumulated through time[8,9]. Furthermore, mountains are home to hundreds of millions of people[10] and provide vital ecosystem services for humans living in these regions and their surroundings (e.g., recreational opportunities, protection against natural hazards,

[1]Division of BioInvasions, Global Change & Macroecology, Department of Botany and Biodiversity Research, University of Vienna, Rennweg 14, AT 1030 Vienna, Austria. [2]Instituto de Ciencias de la Atmósfera y Cambio Climático, Universidad Nacional Autónoma de México, Ciudad Universitaria, Ciudad de México, México. [3]Senckenberg Biodiversity and Climate Research Centre, Frankfurt, Germany. [4]Department of Animal Ecology & Systematics, Justus Liebig University Giessen, Giessen, Germany. [5]Centre of Geographical Studies, Institute of Geography and Spatial Planning, University of Lisbon, Lisbon, Portugal. [6]Associate Laboratory TERRA, Lisbon, Portugal. [7]Instituto Pirenaico de Ecología, CSIC. Avda. Montañana 1005, 50192 Zaragoza, Spain. [8]Division of Biodiversity Dynamics and Conservation, Department of Botany and Biodiversity Research, University of Vienna, Rennweg 14, AT 1030 Vienna, Austria. ✉e-mail: adrian.garcia@univie.ac.at

and regulation of climate, air quality, and water flow; see ref. 11 for a comprehensive review). However, the mountains' vast legacy is not exempt from the increasing pressures of contemporary global change.

In recent years, an increasing number of studies have analyzed the fate of mountains in the context of global change[12]. From a biological point of view, many studies have been oriented to quantify the impacts of climate and land use change on mountain biotas. For example, several studies have shown that species are shifting their distribution range in response to climate change in such regions[13–15], while others have quantified the effects of ice cover retreat[16] and land-cover change[17–19]. However, other drivers of global biodiversity change such as biological invasions have been less studied in the context of mountains or when studied, have largely focused on alien plant invasions[20–23].

As a direct consequence of the growing connectivity and movement of people and goods around the globe, c.37,000 species have already been introduced and established in regions outside their native ranges[24]. Some of these alien species represent serious ecological problems, also affecting economies and human health[25–28]. Worldwide, invasion hotspots across taxonomic groups have been identified on islands and in coastal regions[29]. In the specific case of mountains, biological invasions have been particularly well-documented for alien plants, for example through global initiatives monitoring alien plant invasions and their spread in mountains across continents[30]. Conversely, our knowledge of large-scale patterns of animal invasions in mountains remains limited, since most studies have focused so far on single species and often at local or regional scales. Some examples are the crayfish (*Procambarus clarkii*) in California mountains, wild goats (*Oreamnos americanus*) in Yellowstone National Park, or trout species (e.g. *Salmo trutta* or *Oncorhynchus mykiss*) and alpine marmots (*Marmota marmota*) introduced in the Pyrenees[31–34]. While studies on alien plants in mountains have already provided evidence of the vulnerability of these regions to alien species (see ref. 35 for a detailed review), it is necessary to assess other taxonomic groups to fully understand the threat posed by biological invasions in mountains.

In this study, we focus on five major animal groups (i.e.: freshwater fishes, amphibians, reptiles, birds, and mammals; hereafter vertebrates). We collected georeferenced alien records for these taxonomic groups in mountains worldwide to address the following research questions: 1. What are the spatial patterns of species richness and record density of alien vertebrates in mountains? 2. What are the direction and magnitude of flows of alien vertebrate species reported in mountains between their native and recipient realms? 3. What is the incidence of alien vertebrates in different types of protected areas in mountains? 4. What are the roles of anthropogenic pressures and mountain characteristics in explaining the observed alien vertebrate richness patterns? Here we show that alien vertebrates are widely distributed in mountains, with birds and mammals being the groups with the highest number of species documented in these regions. The most common recipients are the Palearctic, Nearctic, and Australasian, while the Nearctic, Indo-Malay, and Afrotropic realms are the most frequent donors. Concerningly, many of the alien vertebrate species studied have been reported within protected areas in mountainous regions. The distribution of alien vertebrates in mountains is strongly correlated to anthropogenic pressures such as higher road density and lower biodiversity intactness, but also to mountains' physical characteristics, including their elevation range and terrain roughness.

## Results

### Global patterns of distribution of alien vertebrates in mountains

Our final dataset contains 167,357 records of 717 alien vertebrate species (99 fishes, 59 amphibians, 318 birds, 80 reptiles and 161 mammals) from 2978 mountains. Overall alien vertebrate richness in the studied mountains varies between one (in 690 mountains) and 81 species documented specifically in the Cairngorms in United Kingdom (Fig. 1). Along with continental mountains in the Northern Hemisphere and Australia, many mountainous settings in islands like New Zealand (29 spp.), Reunion Island (26 spp.), Canary Islands (21 spp.) and the Lesser Antilles (20 spp. in Trinidad and Tobago) have 20 or more alien vertebrates reported. Accounting for sampling completeness, additional mountains emerged as having higher-than-expected alien vertebrate richness based on the sampling effort conducted in these areas (Supplementary Fig. 5). The mountains with the highest positive deviations (i.e., observed alien richness values way above the expected ones based on the regression between sampling

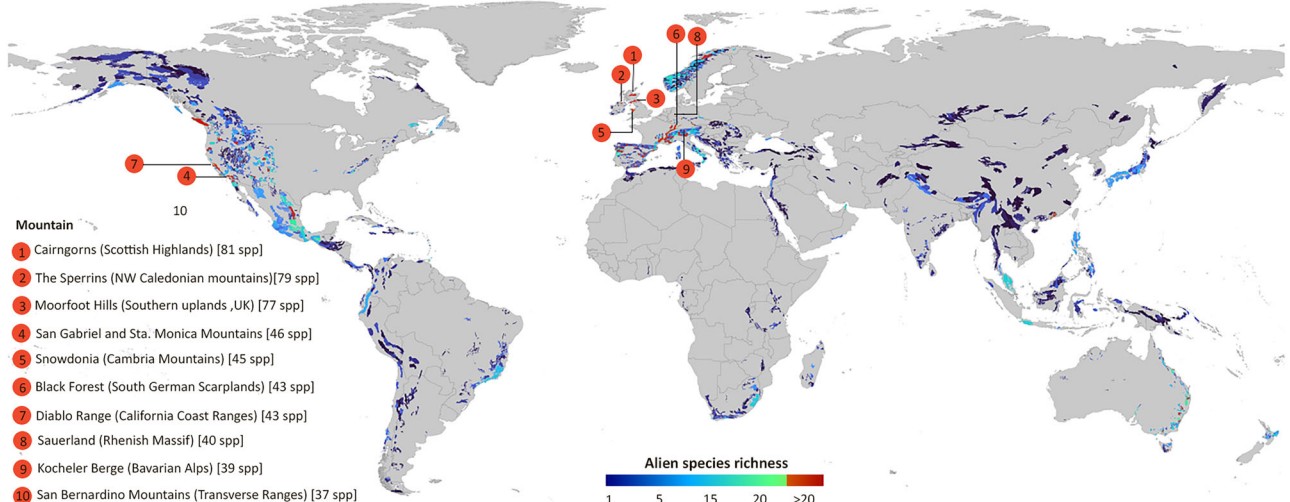

**Fig. 1 | Distribution of alien vertebrate richness across the mountains of the world.** The colored polygons represent the subset of mountains extracted from[72,73] for which our dataset contains records of at least one alien vertebrate species. Highlighted are ten examples of mountains with observed species richness above the 99th percentile of the distribution of values (i.e. >33 species). The legend shows the mountain name, the mountain system to which each mountain belongs,

and the respective number of alien vertebrate species recorded. These mountains remain at the top of the distribution also in the sampling bias-corrected pattern (See Supplementary Fig. 5). Lists of the top ten mountains with the highest number of alien vertebrates overall and per taxonomic group are provided in Supplementary Table 2.

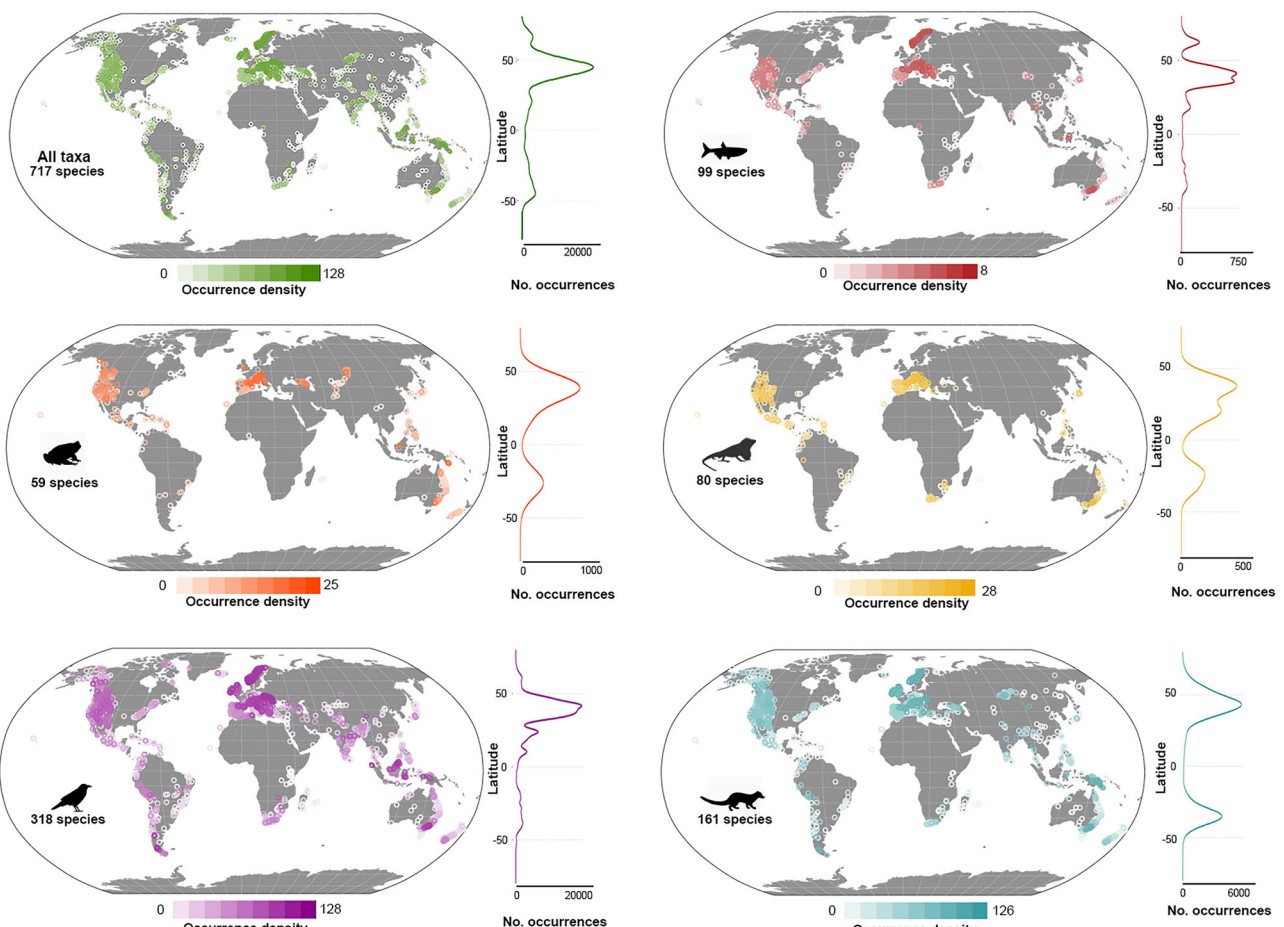

**Fig. 2 | Distribution of alien vertebrate records in mountains of the world** (*n* = 167,357 records of 717 vertebrate species). The panels show the distribution of the overall records compiled (top left), and the distribution patterns of alien species of each vertebrate group studied. The circles in the maps represent the centroids of the mountains with documented alien vertebrates and the color indicates the density of records (i.e., records /area). Density plots on the right side of each map show the latitudinal variation in the absolute number of records for the respective taxon. The silhouettes used in this and the following three figures were obtained from www.phylopic.org/ (*Alburnus alburnus* by Carlos Cano Barbacil, *Rhinella marina* by Dennis C. Murphy; *Anolis carolinensis* by Ingo Braasch; *Corvus corax* by Ferran Sayol and *Genetta genetta* by Pearson Scott Foresman and T. Michael Keesey).

completeness and alien richness) are consistently found in Europe and the US, with residuals ranging between 25 to 70. A few mountains in Mexico, China, Australia, and Brazil have positive residuals above 10 but in all cases below 25 (see Supplementary Fig. 5).

We found that most records of alien vertebrates in mountains correspond to birds and mammals, 127,653 and 29,396 records, respectively, versus the less than 4000 records we compiled for each of the other taxonomic groups. Birds and mammals are also the most widespread groups, reported in 2595 and 1518 mountains across all realms, respectively. On the contrary, amphibians are the group with the fewest alien species reported, distributed across only 458 mountains. All vertebrate groups have records across almost all latitudes, with the number of records peaking at northern temperate latitudes (Fig. 2). However, amphibians, reptiles, and mammals show a second but less pronounced peak of records in subtropical latitudes of the Southern Hemisphere. All groups have low proportions of their total records in low latitudes.

Fishes, amphibians, and reptiles have fewer records in the Neo-tropics (mostly restricted to Central American Highlands, northern Andes, and the Brazilian Atlantic Forest), the Afrotropic, and the Western portion of the Palearctic. Alien amphibians are absent in the Afrotropical mountains, and reptiles are least represented in mountains from the Western Palearctic (Fig. 2).

## Global flows of alien vertebrates occurring in mountains

Flow diagrams show that the overall geographic flows of alien vertebrates in mountains are dominated by the Nearctic, Indo-Malay, and Afrotropical realms as main donors and the Palearctic, Nearctic, and Australasia as main recipients. Nevertheless, the pattern is not consistent across taxonomic groups (Fig. 3). For fishes, the major donors are the Nearctic and the Afrotropic while the Nearctic and the Palearctic are the main recipients. For amphibians the Nearctic and the Palearctic are both main donors and recipients. For birds and reptiles, we identified the Afrotropic and the Indo-Malay realms as the main donors, while the Nearctic and the Palearctic are the main recipient realms. For mammals, the Indo-Malay and Palearctic are the main donors, Australasia and the Palearctic are the main recipients (see detailed species numbers in Supplementary Table 3).

Assessing whether the overall flows of alien vertebrates among realms are higher or lower than expected (see Supplementary Fig. 6) we found that the flows from the Afrotropic to the Palearctic, Oceania and Madagascar are higher than expected by chance considering the native diversity of the donor realm. Similarly, the flows from the Nearctic to the Neotropics, Oceania, and the Palearctic and from the Palearctic to Australasia are also higher than expected. The opposite pattern occurs with the introductions from the Afrotropic to

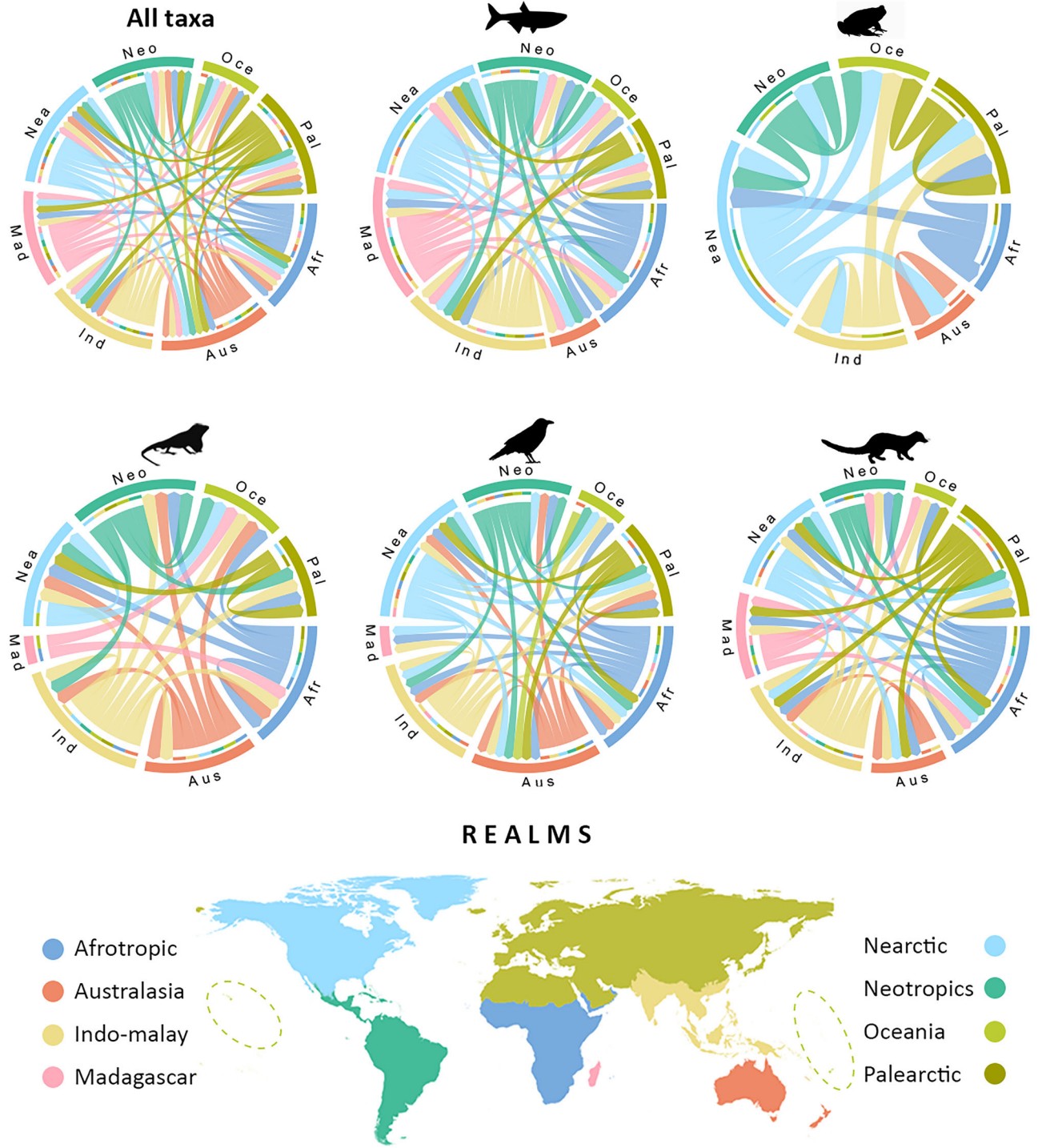

**Fig. 3 | Global flows of alien vertebrates between mountains of different biogeographic realms.** Realms are represented by different colors. The chords show species flows between native range and alien ranges, with broader chords indicating higher species numbers. Segments in the outer circle are proportional to the number of species involved in the flows from (small, colored rectangles) and to (arrows pointing) a determined realm. Biogeographic realms are delimited according to Olson et al.[38]. Given the small size of most territories in Oceania, for reference, we depicted the approximate location of the realm with dashed circles. Antarctica is not shown due to the lack of alien vertebrate records in that realm.

Australasia and the Indo-Malay realms, from Australasia to the Nearctic and Palearctic, and from the Neotropics to all the other realms. In such cases, fewer alien vertebrate species have been transported than expected from the respective available native species pools. Except for Oceania, the intra-realm flows are higher than expected by chance. For example, in the case of amphibians, fishes, and mammals, important flows occur within the Nearctic and the Palearctic.

## Alien vertebrates in protected areas in mountains of the World

We found that 347 alien vertebrate species, represented by 11,230 records (7% of the total records analyzed) occur within 827 PAs in mountains. Only ten species have been exclusively reported within PAs in mountains. Birds (67%) and mammals (26%) again account for over 90% of the alien vertebrate records documented in PAs located in mountains (Fig. 4a). Such records correspond to 179 bird and

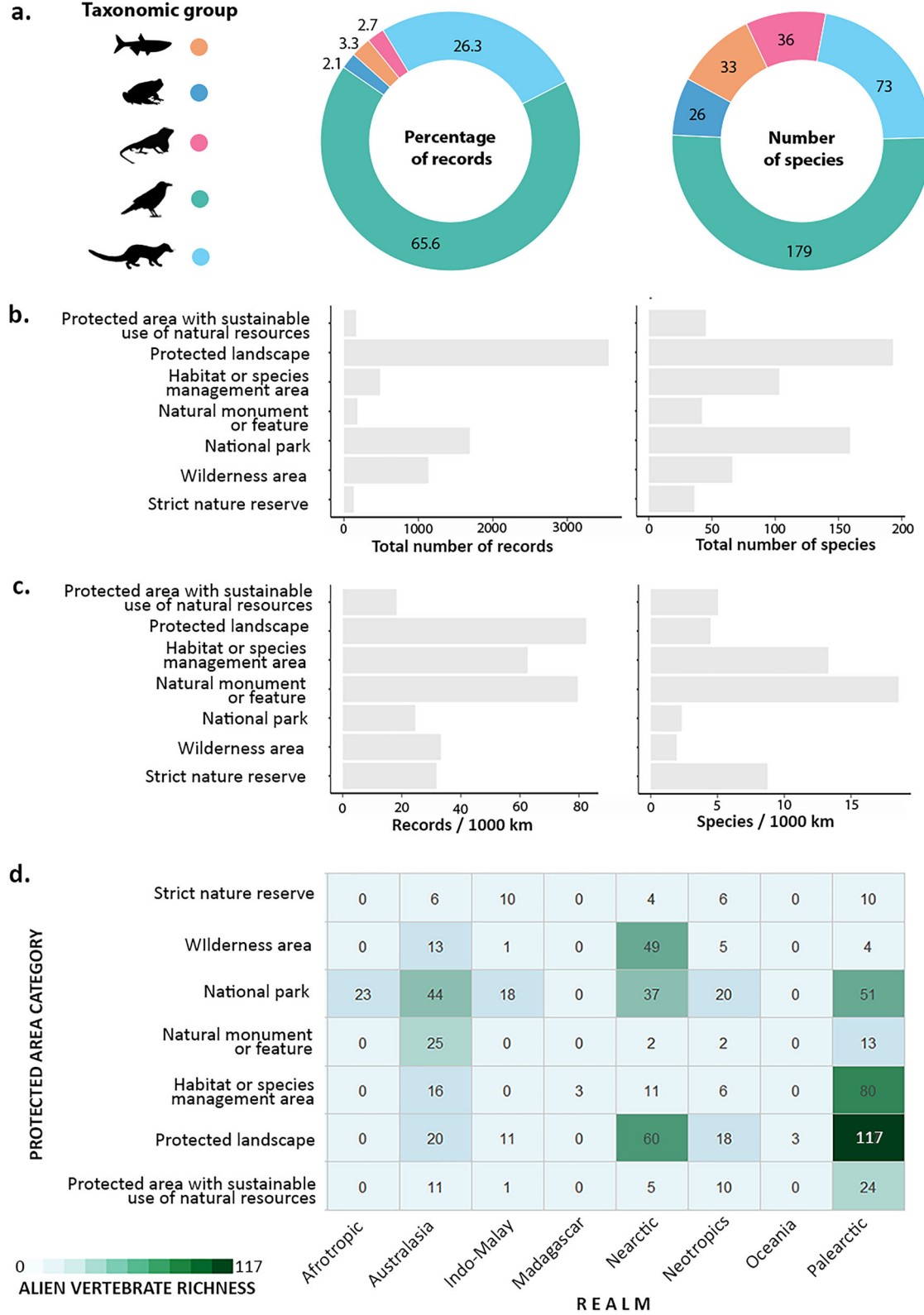

**Fig. 4 | Distribution of alien vertebrates occurring within protected areas (PA) located in mountains of the world. a** Percentage of records corresponding to each taxonomic group and their species richness. **b** Distribution of total records and species richness among the different PA categories and **c** the same statistics but corrected by the area of each PA category. **d** Cumulative alien vertebrate richness found for each PA category within the realms studied.

73 mammal species (70% and 58% of the alien vertebrate species of each group in our dataset) (Fig. 4b). Fish, amphibians, and reptiles, each accounted for less than 4% of the total records detected in mountain PAs. Nevertheless, these few records together represent 95 alien vertebrate species (26 amphibians, 33 fishes and 36 reptiles; Fig. 4a). For the full dataset, most alien records and most species have been documented in Protected Landscapes (4855 records of 243 species) and National Parks (2887 records of 187 species). Strict Nature

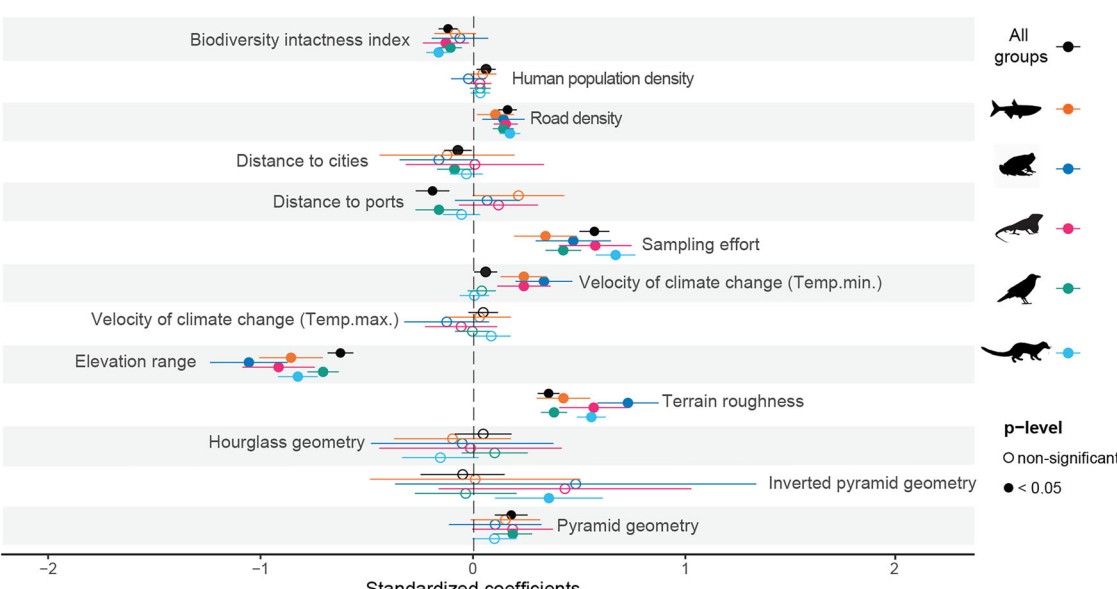

**Fig. 5 | Effect sizes resulting from GLMMs testing predictor variables of alien vertebrate richness in mountains worldwide ($n = 2696$ mountains).** For each predictor the six effect sizes shown represent the different GLMMs fitted: one cross-taxon model (black symbols) and five taxon-specific models, each in a different color. Values to the left of the zero line depict negative relationships between the response variable and the respective predictor (i.e., alien vertebrate richness increases as the predictor decreases), and those to the right show positive relationships. Unfilled circles represent variables with non-significant effects. Filled circles effects with statistical significance ($p < 0.05$). Error bars represent the corresponding 95% confidence intervals. Significances for the geometry types are estimated against the baseline level of a diamond-shaped mountain. For the taxon-specific models we analyzed patterns of alien vertebrate richness variability across the following number of mountains: fishes ($n = 722$), amphibians ($n = 429$), reptiles ($n = 498$), birds ($n = 2424$), and mammals ($n = 1549$).

Reserve is the PA category with the fewest alien records documented and alien species recorded (Fig. 4b). Nevertheless, accounting for the portion of the PAs´ surface located in mountains, Natural Monuments and Habitat/Species management areas also emerge as important categories in terms of records and species number per unit area (Fig. 4c). At the realm level, the PAs with the highest alien vertebrate richness are the Protected Landscapes and the Management Areas of the Palearctic, followed by the Protected landscapes of the Nearctic. National parks have intermediate alien vertebrate richness across most of the realms (Fig. 4d).

**Drivers of alien vertebrate richness in mountains of the world**
The cross-taxon model showed significant positive effects of human population density (estimate [±SE] = 0.057 [0.023], $F_{1,2678} = 2.515$, $p < 0.001$), road density (estimate [±SE] = 0.157 [0.022], $F_{1,2678} = 7.119$, $p < 0.001$), and sampling completeness (estimate [±SE] = 0.562 [0.036], $F_{1,2678} = 15.810$, $p < 0.001$) on the distribution of alien vertebrates in mountains of the world. We also found positive effects of terrain roughness (estimate [±SE] = 0.348 [0.026], $F_{1,2678} = 13.403$, $p < 0.001$) pyramid geometry (estimate [±SE] = 0.174 [0.039], $F_{1,2678} = 4.465$, $p < 0.001$) and velocity of change of minimum temperature (estimate [±SE] = 0.055 [0.027], $F_{1,2678} = 2.014$, $p < 0.05$). Biodiversity intactness (estimate [±SE] = -0.119 [0.023], $F_{1,2678} = -5.306$, $p < 0.001$), distance to cities (estimate [±SE] = -0.074 [0.033], $F_{1,2678} = -2.255$, $p < 0.05$) and ports (estimate [±SE] = -0.192 [0.040], $F_{1,2679} = -4.788$, $p < 0.001$) and elevation range (estimate [±SE]= -0.620 [0.031], $F_{1,2678} = -20.206$, $p < 0.001$) on the other hand have negative effects (Fig. 5).

From the taxon-specific models, we found that driver effects vary among groups. Alien richness of all vertebrate groups increases with higher road density, higher sampling completeness, narrower elevation range, and rougher topography (Fig. 4). The degree of BII inversely affects reptiles (estimate [±SE] = -0.129 [0.055], $F_{1,480} = -2.360$, $p < 0.05$), birds (estimate = -0.109 [0.027], $F_{1,2406} = -4.009$, $p < 0.001$) and mammals (estimate = -0.164 [0 .029], $F_{1,1531} = -5.587$, $p < 0.001$). The velocity of climate change regarding minimum temperature is relevant to explain the distribution of alien fish (estimate = 0.233 [0.055], $F_{1,704} = 4.235$, $p < 0.001$), amphibians (estimate = 0.327 [0.068], $F_{1,411} = 4.838$, $p < 0.001$) and reptiles (estimate = 0.233 [0.063], $F_{1,480} = 3.679$ $p < 0.001$) in mountains (Fig. 5). Sensitivity analyses fitting models with different subsets of mountains showed consistent results overall (Supplementary Fig. 7).

## Discussion

By compiling and analyzing data on over 700 vertebrate species reported as having alien populations in mountains worldwide, we showed that in nearly 3000 mountains across the globe, at least one vertebrate species has been documented. Recorded richness of alien vertebrates, peaks in mountains of the Palearctic and Nearctic and the highest number of species are native to the Nearctic, the Indo-Malay, and the Afrotropical realms. Many alien vertebrates have also been reported in PAs within mountain regions, mainly in areas under the categories of Protected Landscapes and National Parks. The major drivers of alien vertebrate richness in mountains have been identified as being related to human activities but also to mountain intrinsic features.

### Patterns of distribution of alien vertebrates in mountains worldwide

Alien vertebrate richness peaks in Northwestern European mountains particularly in the United Kingdom. Several other mountains in Europe show high richness in countries like Germany, Switzerland, and France. Likewise, mountains on the Pacific Coast of the United States, in Australia and New Zealand are also notable for the high numbers of alien vertebrate species (see Fig. 1). This general pattern does not differ much from the global picture of established alien species richness (considering both animal and plant groups), for which many hotspots emerge in temperate latitudes[29]. Islands have been also highlighted as hotspots of alien species worldwide[29]; here, we found that this is also consistent for alien vertebrates on islands with mountainous settings across the globe (e.g., New Zealand, Reunion Island, Canary Islands or Trinidad and Tobago). This is particularly concerning given the known

role of biological invasions as a main driver of extinction on islands worldwide[36,37].

In high-latitude mountains, we also found the highest density of alien vertebrate records, a pattern that stands consistently among groups (Fig. 1). This fact echoes the known existing bias in sampling completeness among different regions[38], but also across elevation gradients, where high elevations are particularly under-sampled in terms of biodiversity data in general[39]. Consequently, we stress the need for further efforts to reduce these gaps, which are especially disproportionate in tropical regions, as has been documented also for other taxonomic groups[40]. For vertebrates, we found few to no records of alien amphibians in regions like the Andes and most African mountains, and the same for fishes and reptiles in the mountains of mainland Asia. Disentangling whether this mirrors a lack of threat in such regions or an underestimation due to low sampling requires more research. On the one hand, an increase in monitoring initiatives is necessary to document new records, but also a targeted effort to retrieve and mobilize published information from the non-English-language scientific literature is an important first step to take[41].

Regarding the large differences in total records available between birds and mammals relative to fishes, amphibians, and reptiles, an additional factor to consider is the variation in the probability of species detection among groups, which in turn might be influenced by additional species intrinsic factors or even survey methods implemented[42]. For instance, small body sizes, nocturnal habits, or inconspicuousness could impose major limitations on the proper identification of some species. Such constraints could be even more significant in citizen science initiatives that, on the other hand, are very successful in generating, for example, disproportionate amounts of bird observations[43].

## Flows of alien vertebrates among realms

As expected from the global patterns of distribution of alien vertebrates in mountains described above, the Palearctic and the Nearctic are the main recipient realms. Interestingly, along with the Nearctic (244 species), the Indo-Malay (241 species) and the Afrotropical (222 species) realms are the predominant donors. This implies an important flow from the southern to the northern hemisphere. One example are the records of 70 Afrotropical vertebrate species that have been documented in Palearctic mountains, which in turn is higher than expected based on the Afrotropical pool of native vertebrates. Using a similar approach, but evaluating global flows of alien plant species, previous studies[44,45] have found that the Northern Hemisphere continents have been the major donors to all other continents. Instead, we found a higher-than-expected northern to the southern hemisphere flow pattern only for Nearctic vertebrates ($n = 32$) recorded in Neotropical mountains. Variable intensity in the flows between the southern and northern hemispheres could be partially determined by specific introduction pathways that are imposed by activities such as the pet market, which is in turn influenced by cultural differences in pet-keeping traditions across regions[46].

At the class level, we found differences in the major flows of species among and within realms. For amphibians, fishes, and mammals, main flows occur within the Nearctic and the Palearctic. Our findings are supported by previously observed dominant intra-continental flows of established amphibian species, in North America and Europe[47]. In the case of freshwater alien fishes, recent analyses of global flows also highlight the Palearctic as the top region experiencing the largest internal flows[48]. Conversely, the within-realm flow that we detected in the Nearctic is not found in established fish species, for which other regions (e.g., the Sino-Oriental zoogeographic realm) are more important[48]. For mammals, it is known that cooler regions have a higher established alien mammal richness[29]. In our data, this is best reflected by the within-region flows of mammals (39 and 29 species exchanged within the Palearctic and the Nearctic, respectively).

For reptiles, instead, we found that predominant flows took again place within the Palearctic but also within the Neotropics, closely followed by the internal exchange in Australasia. Our results align well with the global patterns previously found for this group, as these global flows are rather balanced with important recipients and donors in multiple regions[47]. Birds are the only group where an inter-realm flow was predominant, specifically the 56 Afrotropical alien species documented in Palearctic mountains, but we also found important species fluxes within the Australasian (46 spp.) and the Indo-Malayan (45 spp.). The specific Afrotropical-Palearctic flow in recent decades was reduced due to the European Union ban on imports of wild-caught birds[49], but its historical signature is still evident when looking at our data. As in the case of birds, global wildlife trade is among the most important means of spread of vertebrates worldwide, affecting over 7500 species of terrestrial vertebrates[50,51] and likely also influencing the patterns we found in mountains.

## Alien vertebrates in mountain protected areas

A global analysis recently revealed that less than 10% of the PAs of the world are home to alien animal species. Nevertheless, the study shows that there is at least one established population within 10-100 km of the boundaries of almost any PA worldwide[52]. In the case of mountains, we found records of alien vertebrates within the boundaries of more than 820 PAs globally. As expected, birds and mammals -groups largely overrepresented in our dataset- have also the highest number of records and the highest alien species richness. Interestingly, although only 7% of all records refer to the other three taxonomic groups, these 7% represent more than one-quarter of the species reported in PAs in mountain regions.

Regarding the type of PAs where alien vertebrates have been documented in mountains, in absolute numbers most records and the highest alien species richness are in Protected landscapes and National Parks. Both categories are characterized by a high interaction of people with nature, whether through recreational, scientific, or daily activities. In Strict Nature Reserves, where such visitation is minimal, the total records and number of alien species are reduced. Previous studies have shown that, at least in Europe, human accessibility is a major predictor of alien species richness in PAs[53]. Correcting for the area, our data shows that this trend is even more evident as small and accessible protected areas show the highest numbers of both records and species richness relative to their area. While accessibility and human population density have proven to be good predictors of detection probability in different taxonomic groups[54–56], it is still an open question what the relative contribution of human presence is in increasing both the detection probability and the introduction rates of alien species.

## Drivers of alien vertebrate richness in mountains

We found that for most taxonomic groups, mountains characterized by a lower biodiversity intactness index, more developed road networks, closer proximity to ports and cities, more complete sampling efforts, as well as those experiencing rapid recent increase in minimum temperatures tend to have a higher overall number of alien vertebrate species. These predictors are directly linked to anthropogenic pressures and add to the increasing evidence of the strong influence of human footprint in driving the spatial reconfiguration of life on Earth[57]. Moreover, several mountain physical characteristics, like pyramidal geometries and narrower - but topographically complex - altitudinal ranges are also good predictors of alien vertebrate richness. Larger available areas in the lower elevational belts are more accessible for a higher number of species -as seen in plants, for instance[58]. Moreover, the heterogeneous mosaic of conditions associated with rough terrains may provide suitable settings for the persistence of many alien vertebrates. While broad elevational ranges can create diverse climatic conditions, offering suitable environments that facilitate the

establishment of multiple species, they can also present significant physical and physiological barriers, limiting the dispersal and survival of others[59].

By deconstructing the general picture into models separately fitted for each vertebrate group (i.e. fishes, amphibians, reptiles, birds and mammals), we found that the effects of the tested drivers vary across different taxonomic groups. This variation may be influenced by species intrinsic traits, such as particular reproductive strategies or the different natural dispersal abilities after initial introductions[60]; unfortunately, we could not consider life history traits due to the limited data availability. Nevertheless, we may expect that overall, most groups will continue spreading rapidly in the upcoming years. Recent evidence shows that the secondary spread of alien species is much faster than the spread of native ones and the velocity of climate change[61]. This may lead to the replacement of native species by alien ones, likely resulting in strong ecological repercussions in vulnerable regions like mountain ranges. Since these regions are characterized by high endemism rates and many native species have very restricted distribution ranges[62], such turnover will likely lead to local extinctions and biotic homogenization as demonstrated in other taxonomic groups[63].

Regarding the common drivers among groups, we found that the alien richness of all vertebrate groups increases in mountains with higher road density, higher sampling completeness, narrower elevation ranges, and rougher terrains. These results suggest that independent of the biological particularities of each group, the emerging richness patterns of alien vertebrates in mountains are strongly tied to infrastructures that promote connectivity and facilitate the human-driven spread of introduced species, but also to sampling efforts, and heterogeneous landscapes. Supporting this, previous monitoring efforts following standardized protocols[30] have revealed that -at least for plants[64]- roads provide favorable habitat and anthropogenic dispersal routes for many alien species in mountains (reviewed in[65]). Moreover, our findings indicate that as sampling efforts improve in understudied regions, the detection of alien vertebrates may increase, particularly in areas with rugged terrain, where the diverse environmental conditions could offer more opportunities for introduced species to establish.

Other predictors linked to global change were important for specific groups. For example, the local increase in minimum temperatures during the last century was a significant and positive predictor for alien ectotherms (i.e. fishes, amphibians, and reptiles). This suggests that rising minimum temperatures due to anthropogenic emissions could enable the expansion of these groups into higher elevations, which is particularly concerning considering that the climate in mountains is expected to change three times faster than the global average[66]. Evidence of altitudinal range shifts due to climate change has been documented in various taxonomic groups within their native ranges[67]. However, our findings suggest that upslope shifts could also be possible for ectotherms in non-native regions. In these cases, human-facilitated introductions could substantially contribute to the spread of species that otherwise would be limited by their relatively poor dispersal abilities compared to birds and mammals. These idiosyncratic responses also make evident the degree of complexity of the ongoing global changes triggered by multiple and interacting human pressures that threaten mountain biodiversity.

Our study provides a comprehensive synthesis that, for the first time, presents a global overview of the current situation of alien vertebrates in mountainous regions. While the scale and resolution of our work precludes us from making specific recommendations for individual mountains or species, the patterns we identified underscore critical need of keeping mountains as pristine as possible to minimize the spread of alien vertebrates. To achieve this, efforts should focus on reducing the human footprint in these areas, where it is increasing due to factors such as rising tourism, as well as novel anthropogenic

pressures, for example the ones linked to the accelerated development of renewable energy infrastructure[68]. Similarly, restrictions on the construction of new roads and the development of hiking trails in mountainous regions should also be considered, as these can serve as potential pathways for alien species[69,70]. This is especially relevant for tropical mountains which have been historically underrepresented in the global protected area network[71] and for mountains in general, as climate change may relief cold-temperature constraints potentially triggering a boost of additional invasions into mountain areas in the near future. We hope that our global-scale analysis will stimulate intensified research on mountain invasions to guide the conservation of these peculiar ecosystems.

## Methods

### Geographic delimitation of mountain ranges

To map global mountains, we used the most up-to-date inventory of global mountains, which contains more than 8500 mountain ranges[72,73]. Compared to previous definitions of mountains that relied on expert opinion (e.g.,[74]), this inventory is built based on parameters derived directly from digital elevation models (DEMs) and implements an approach that provides high accuracy, using, for example, rivers to establish borders between contiguous mountain ranges. From this dataset (GMBA Inventory v.2.0 standard), we only considered non-overlapping mountain polygons classified as "Mountains with well-recognized names" at the most basic mapping unit (i.e. mountains without smaller subdivisions) as our study units ($n = 4953$ mountains across the globe). This level of classification allows us to describe in detail the alien distributions and richness patterns in the mountains of the world and analyze their underlying drivers. Moreover, given the hierarchical structure of the dataset, these units can be grouped into higher categories, which allows assessing the influence of different geographic scales on spatial biodiversity patterns. Note that mountains are defined based on the based on the roughness of the landscape, and not by elevation, therefore many may also include adjacent lowlands when they are rough enough. The distribution of lower and upper elevation limits covered by the studied mountains and a sample of their variable elevation ranges is shown in Supplementary Fig. 1. The latitudinal distributions of these mountain parameters are shown in Supplementary Fig. 2.

### Study species and their alien records in mountains

We based our analyses on the recently published DASCO database[75], which provides geographic details for over 35 million alien records of plants and animals worldwide. DASCO data is derived from point-wise records extracted from GBIF and OBIS that have been cleaned and categorized as alien based on regional invasion checklists (see details on coordinate cleaning in ref. 75). It is important to acknowledge that GBIF, and consequently DASCO, do not explicitly discriminate between casual and established self-sustaining populations, and therefore they might overestimate the current distribution range of alien species. Following the precautionary principle, we analyzed DASCO's records, preferring overestimation to underestimation. This approach reduces the risk of incomplete descriptions of alien species spread by also considering casual populations that later may exert impacts on biodiversity and ecosystem services.

We first subset DASCO's data to retain only the records belonging to vertebrate species. We then filtered this subset to keep only those records located within the above-described polygons of mountains. Since DASCO is mostly based on country-level checklists, some records may be misclassified, for example, when a portion of a given species distribution is reported as alien only in a region of a given country but is native in the rest. Therefore, we double-checked the alien status of each record by using a spatially explicit filtering procedure that integrates available geographic information on species' native ranges. To perform this, we obtained distribution range

polygons for fishes, amphibians, reptiles, and mammals, from IUCN Red List Data (http://www.iucnredlist.org/, downloaded on June 8, 2023), and for birds, we used the range maps available at eBird (www.eBird.org). Then, we subset the shape maps to retain only the polygons depicting the native distribution of the species. For each species, we then discarded from the DASCO's vertebrate records' subset, those records falling within the native range polygons. After this procedure, we ended up with 167,357 alien records of 717 species - 99 fishes, 59 amphibians, 318 birds, 80 reptiles, and 161 mammals - and spanning 2953 different mountains across the globe. To visualize the global distribution of alien vertebrate records in mountains, we plotted in the geographic centroid of each mountain the value of alien vertebrate richness accounting for the mountain area (extracted directly from[72]). For this, we estimated the values of alien record density by dividing the number of total alien records and those of each taxonomic group reported per mountain with the respective mountain area. Then we mapped this metric for all vertebrates and taxonomic groups separately. We implemented the steps of this procedure using the R packages *sp* (v2.1)[76], *sf* (v1.0)[77] and *terra* (v1.7)[78].

### Global flows of alien vertebrates in mountains

We assessed the global flows (i.e., species displacements from native to alien regions) of alien vertebrates in mountains by measuring species flows between biogeographic realms. For this, we used Olson's biogeographic realm classification[79]. This classification consists of nine realms that represent units of largely distinct biotas. For each taxonomic group, we first filtered the polygons depicting native distributions and used them to extract the species' native realms using the Olson's et al. [79] delimitation[79]. Similarly, for each alien record in our dataset, we assign the realm in which they occur by overlapping the realms and the alien records layer. Using this information, we quantified the number of species moving from each native realm to the alien ones for each taxonomic group. We then used this input to depict the flows between native and invaded realms for each species using the R package *circlize* (v0.4.16)[80].

To test whether the flows observed among native and invaded realms are higher or lower than expected by chance, we created a series of null models. We first compiled information on the native realms of roughly 38,000 vertebrate species, to create a global pool of vertebrates, based on the range maps from IUCN and eBird used in the previous sections. Then, we randomized the alien vertebrate species composition of each realm by resampling from the full pool of global vertebrates the number of species (with their respective native realm) documented as having alien records in each realm. We repeated this procedure 999 times to generate a random distribution of simulated alien compositions on each realm, which we then compared with the observed one to assess the statistical significance of differences. The observed number was considered smaller or greater than expected when it was in or beyond the lower 2.5% or upper 2.5% of the distributions of the 999 random draws, respectively.

### Alien vertebrates in protected areas in the mountains of the world

To assess the incidence of alien vertebrate records in mountain protected areas (PAs), we used the World Database on Protected Areas (WDPA, available at www.protectedplanet.net/, downloaded on September 5, 2022). WDPA is an up-to-date source of over 260,000 PAs globally (UNEP-WCMC & IUCN, 2020). Following established protocols[52,81], we only considered PAs classified as "designated", "inscribed" and "established", and having an assigned IUCN Protected Area category, in our analysis. We overlapped our alien records over the polygons depicting such PAs, to extract information on the PA category available from the WDPA layer (see Supplementary Table 1 for a full definition of the IUCN PA categories considered). Then, we calculated the relative contribution of each taxonomic group to the total

records identified as occurring within any PA category. Similarly, we calculated the alien species richness of each taxonomic group occurring in PAs across the mountains of the world by summing up the taxa with records documented within PAs. We also quantified the total number of alien records and the total number of alien vertebrate species occurring within each PA category. This procedure was then repeated accounting for the surface area of the studied PA's, considering only the portion of surface located within mountains. Finally, we used information on realm delimitation to identify the number of alien vertebrate species occurring within each PA category at the realm level.

### Predictors of global patterns of alien vertebrate richness in mountains

To investigate the drivers shaping the global patterns of alien vertebrate richness in mountains, we compiled data on nine predictor variables describing the human footprint in mountains as well as mountain physical characteristics: 1. Human population density, 2. Road density, 3. Distance to cities, 4. Distance to ports, 5. Biodiversity Intactness, 6. Velocity of climate change, 7. Terrain Roughness 8. Mountain geometry, 9. Elevation range, and 10. Sampling completeness. Most of these variables were compiled from existing global maps, metrics estimated here are described in detail below and all data sources used are available in Table 1.

Averages for human population density, road density[82], and biodiversity intactness[83] were calculated after extracting all the respective variable values for each mountain using the polygons delimiting mountain boundaries. In the case of distance to ports and cities[84], we extracted the minimum value representing the nearest distance to settlements with >50,000 inhabitants and the nearest distance to ports. We calculated the gradient-based velocity of climate change (gVoCC)[85]; over a high-resolution time series of monthly minimum and maximum temperatures from 1900 to 2018 from the CRU-TS database[86]. The gVoCC captures the speed of climate conditions for a given cell based on the neighboring cells and was estimated using functions from the *VoCC* R package (v1.0.0)[87].

To quantify topographic complexity, we calculated terrain roughness, a metric defined by the difference between the maximum and the minimum elevation of a cell and its eight surrounding cells[88]. We first used a Shuttle Radar Topography Mission elevation layer at 30 secs resolution (SRTM[89]) to create a global map of roughness and extracted all overlapping cell values to estimate the mean value of roughness per mountain. To determine mountain geometry, we followed the approach proposed by ref. 90 to categorize each mountain into one of four classes: pyramid, inverse pyramid, diamond, or hourglass. This categorization is based on the skewness and modality of the distribution of elevations across the entire mountain range. We estimated skewness using the R package *moments* (v 0.14.1)[91], for modality we applied a Hartigan's dip test for each mountain. We assign the hourglass class to all those mountain ranges with dip values > 0.01 and significant deviations from unimodality (p < 0.05), irrespective of skewness. For the remaining mountain ranges, we assigned the pyramid class to those having skewness ≥ 0.5. The inverse pyramid class was assigned to those with skewness ≤ −0.5. Mountain ranges with skewness values between −0.5 and 0.5 were assigned to the diamond category. In other words, pyramidal mountains have more available area in lower elevations and show a linear decrease of area with the increase in elevation, inverse pyramids display the opposite pattern. In diamond mountains, the available area peaks at intermediate elevations, whereas in hourglass mountains there is more available area in lower and higher elevations, showing a bimodal distribution of elevations.

We used available data on inventory completeness for amphibians, birds, and mammals available from[38]. This variable is calculated as the difference between the expert estimation of species richness

**Table 1 | Potential drivers of alien vertebrate richness in mountains of the world**

| Predictor | Description | Source | Rationale |
|---|---|---|---|
| Human population density | Average population density (number of individuals per km$^2$ at 2020) from values extracted for each mountain. | Layer available at: https://beta.sedac.ciesin.columbia.edu | Human population size positively influences the chances of species introduction and detection. |
| Road density | Average amount of meters of road per km per mountain, considering all road types. | Layer available at: https://www.globio.info See[82]. | Road connectivity increases transport and therefore likelihood of the spread of alien species. |
| Distance to cities | Minimum distance (in km) to settlements with > 50k inhabitants | Layer available at: https://doi.org/10.6084/m9.figshare.7638134.v3 See[84]. | Proximity to human settlements increases the introduction and detection rate of alien species. |
| Distance to ports | Minimum distance (in km) to the nearest port | Nelson et al. [84]. Layer available at: https://doi.org/10.6084/m9.figshare.7638134.v3 See[84]. | Alien richness increases in the surrounding areas of ports due to their role as hubs of introduction. |
| Biodiversity Intactness Index (BII) | Average abundance of native terrestrial species in a region compared with their abundances before pronounced human impacts | Layer available at: https://data.nhm.ac.uk See[83] | Low BII regions are more affected by human intervention, including the introduction of alien species. |
| Velocity of climate change | Gradient-based velocity of climate change (gVoCC, km/year), a temporal climatic trend ratio at a given grid cell to the spatial climatic gradient across a given neighborhood of cells | Estimated from the CRU-TS dataset, following[85] and[99] | Contemporary climate warming facilitates the colonization of wider altitudinal gradients by alien species adapted to warm environments. |
| Terrain roughness | Difference between the maximum and minimum elevation of a cell and its eight surrounding cells- | Estimated from an SRTM layer 1 km resolution. (https://www.worldclim.org) | Rough terrains provide a wider suite of habitats that could be occupied by larger number of alien species |
| Mountain geometry | Pyramid, inverse pyramid, hourglass, or diamond geometry depending on the skewness and modality of the distribution of elevations. | Estimated from an SRTM layer 1 km resolution. (https://www.worldclim.org) Following[90]. | Mountains with larger available areas in lower and more accessible elevations support higher alien species richness |
| Elevation range | Upper minus lower elevation limits of each mountain | Calculated here from data available in ref. 72 | Heterogeneous conditions in wider elevation ranges support higher alien richness |
| Sampling completeness | Completeness of native species inventories | Calculated here based on data from[38] | The chances of detection and reporting alien species increases with sampling completeness |

Description, sources, and rationale for the inclusion of the variables describing anthropogenic impacts on mountains as well as physical and ecological mountain descriptors[82].

and the actual species richness recorded by GBIF. We rasterized centroid estimations of inventory completeness for each taxonomic group and then averaged them to obtain a map with the global distribution of mean sampling completeness. From this layer, we extracted the values for all grid cells at 1-degree resolution located within each mountain polygon and calculated mean sampling completeness as a proxy of sampling bias. This approach has been successfully adopted in other recent studies on alien species[92,93].

### Statistical analysis

To identify the existing relationships between the estimated predictor variables and the variation in alien vertebrate richness across mountains, we fitted cross-artaxonomic and vertebrate group Generalized Linear Mixed Models (GLMMs), including the ten explanatory variables as fixed effects. Numeric variables (i.e., all but mountain geometry) were assessed for multicollinearity by estimating Pearson correlations among all of them, to exclude highly collinear variables (i.e. those with absolute r > 0.7) from the GLMMs following[94] (Supplementary Fig. 3). The variables were then scaled to mean = 0 and SD = 1 to enable a direct comparison of the predictor importance. In the case of mountain geometry, which is a categorical variable, we chose the diamond shape as a reference to estimate the significance of the other three geometry types. We included mountain systems nested in regions and these nested in continents as a random effect in our models. Moreover, given the large variability in areas among mountains and the well-known nonlinear nature of species-area relationships[95], we included a log-transformed area as an offset in the model. We obtained the mountain area for each of the study units directly from the mountain dataset[72]. We fitted the models using a negative binomial distribution to deal with overdispersion in our data. The models were implemented using the R package lme4 (v1.1-14)[96]. Additionally, we used the mgcv R package (v1.9-1)[97] to fit an overall Generalized Additive Mixed Model (GAMM) using the same model structure. With this model, we assessed whether the predictors have strong non-linear relationships with our response variable. Since this was not the case, and the output of both approaches was similar (Supplementary Fig. 4), we based our discussion on the results from the GLMMs to facilitate the interpretation of the direction and magnitude of the significant effects found.

### Reporting summary

Further information on research design is available in the Nature Portfolio Reporting Summary linked to this article.

## Data availability

Raw occurrence data analyzed in this study was obtained from the DASCO dataset available at https://zenodo.org/records/10054162. The specific sources of the data used to calculate the predictor variables evaluated are specified in Table 1. The information compiled to run the main analyses performed in this study are available at the Github link provided below.

## Code availability

Codes and final datasets needed to run the analyses conducted in this study are available at https://github.com/Garcia-Rodriguez/Alien-Vertebrates-in-Mountains and have been also deposited in Zenodo[98].

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

## Acknowledgements

A.G.R., B.L., A.O., and F.E. appreciate funding from Austrian Science Fund FWF (Global Plant Invasions, grant no. I 5825-B). A.S. appreciates funding from Austrian Science Foundation FWF (grant no. P 34688-B). JAV is grateful to the Programa de Investigación en Cambio Climático (PINCC) de la UNAM for support through a grant. B.G. received funding from the European Union through the Life-SIP European Program (LIFE22-IPC-ES-LIFE-PYRENEES4CLIMA/101104957). HS acknowledges funding by the Deutsche Forschungsgemeinschaft (DFG, German Research Foundation) (grant no. 521529463). This research was funded in whole or in part by the Austrian Science Fund (FWF) [DOI: 10.55776/I5825, available via https://www.fwf.ac.at/en/discover/research-radar]. For open access purposes, the author has applied a CC BY public copyright license to any author accepted manuscript version arising from this submission.

## Author contributions

A.G.R., B.L. and F.E. designed the study. Analyses were performed by A.G.R with the contribution of J.A.V and enriching feedback from B.L., S.D, A.S., H.S., B.G., C.C and A.O. A.G.R led the writing with significant input from all co-authors who contributed to preliminary drafts and approved the final version of the manuscript.

## Competing interests

The authors declare no competing interests.
