## [Transparent Peer Review file · Nature Communications]

The global distribution patterns of alien vertebrate richness in mountains

Corresponding Author: Dr Adrián García-Rodríguez

Version 0:

Reviewer comments:

Reviewer #1

(Remarks to the Author)

Mountain areas are conservation hotspot of native biodiversity. However, we still understand little on the non-native species invasions in mountain areas at the global scale. This study used global dataset to analyze the distribution, flow, pattern in protected areas, and the drivers' of alien vertebrate species in mountains worldwide. So, this is a novel topic in invasion ecology. I provide some concerns and suggestions to further strengthen this manuscript.

Line 40-51: For the first paragraph of the Introduction section, in addition to the importance of mountains in biodiversity conservation and providing ecosystem services, please also address the environmental change challenges the mountains are facing in the current era of global change.

Line 64-69: Although it is true that it remains little investigated on large-scale patterns of animal invasions in mountains, please provide more details on the invasion cases of alien animals at the local and regional scales.

Line 74: Please clarify what does "dispersal dynamics" mean here? How to confirm the spread stage along the introduction-establishment-dispersal invasion process?

Line 75: How to determine the independent role of PAs in resisting or buffering (the authors wrote here) biological invasions by controlling for the potential effects of human activities?

Line 97-110: As mountains are usually located in some natural regions far from urban areas, there might be literatures reported the occurrence of established non-native species using non-English-languages, which are needed to be included in the data collection (e.g., Amano et al. 2021).

Ref. Amano, T., V. Berdejo-Espinola, A. P. Christie et al. 2021. Tapping into non-English-language science for the conservation of global biodiversity. *PLoS Biology* 19: e3001296.

Line 97-121: My major concern on the non-native species data is that the authors did not validate the establishment status of each non-native species. Compared with only non-native species occurrence data, it should confirm that whether these non-native species have established feral populations in mountain areas.

Line 155: How about the invasion meltdown effect? It is needed to include the number of prior presence of other established non-native species (e.g., Redding et al. 2019).

Ref. Redding, D. W., A. L. Pigot, E. E. Dyer, Ç. H. Şekercioğlu, S. Kark, and T. M. Blackburn. 2019. Location-level processes drive the establishment of alien bird populations worldwide. *Nature* 571:103-106.

Line 172: It is not clear that whether the topographic complexity has been incorporated into the climate change velocity (e.g., Sandel et al. 2011).

Ref. Sandel, B., L. Arge, B. Dalsgaard, et al. 2011. The influence of Late Quaternary climate-change velocity on species endemism. *Science* 334:660-664.

Line 197-198: For the species completeness variable, is it downloaded from an existing database or re-calculated in this study? Please clarify the data source information and provide the R code to generate this important variable.

Line 207: How about the rationale of the correlation value < 0.75 ?

Line 209: It is better to use an AIC-based model averaging method to calculate the relative importance of each predictor variable.

Line 210: It is not clear how the 'diamond shape' was quantified here.

Line 212: In addition to the mountain systems nested in regions and in continents, it should also include the taxonomic identity of different taxa used in the analysis to account for the taxonomic sample non-independence.

Line 221-249: As the authors have been able to account for the sample bias issue as mentioned in the Method section, it should be reported the bias-controlled pattern of distribution of established non-native vertebrates in global mountains.

Additionally, does the observed non-native species altitudinal pattern reflect the classic mid-elevation peak pattern (i.e., mid-

domain effect) of native species (e.g., Quintero and Jetz 2018) ?

Ref. Quintero, I., and W. Jetz. 2018. Global elevational diversity and diversification of birds. *Nature* 555:246-250.

Line 309-328: How about the relative importance of each predictor variable in explaining the established non-native vertebrate richness in global mountains?

Line 350: In addition to these similarities with the global picture of non-native vertebrates, are there any differences observed in the mountain areas compared with other areas from the present study?

Line 359: So, it would be interesting to see a sampling bias-controlled global pattern of established non-native vertebrates.

Line 443-452: After the initial introductions by humans, the subsequent spread of the established non-native species may also be highly related with the natural dispersal abilities among taxa, which are unfortunately lacking here.

Finally, as mountains have rich endemic species just as what the authors have mentioned in the Introduction section, the authors need strengthen the Discussion by addressing the potential impacts of the established non-native vertebrate on mountain endemic species.

(Remarks on code availability)

Reviewer #2

(Remarks to the Author)

Research on alien species in mountains is largely biased towards plants. Accordingly, by looking at alien vertebrate species in mountains this paper contributes to filling a gap and is an important contribution to our understanding of the threat that invasive alien species represent for mountain ecosystems. As such I very much welcome this effort that I see as important.

The authors make smart use of existing data on alien species and map them onto mountain ranges to detect spatial patterns in species richness, including in PAs. Additional analyses generate a number of results on the flow of species and on possible drivers.

Besides individual and relatively small comments to the introduction, methods, and result (yet some additional clarifications are needed in the method section) one of my main concern with this paper is that the discussion is from my point of view very/too shallow as it offers very few explanations for the obtained results and hardly informs alien species research, management, and policy in mountains. Accordingly, whereas the work per se is certainly useful and necessary, the significance is not fully achieved as I remain with the question of "and so what", "where do we go from there". I believe that additional analyses (e.g. by species subgroups, or that look at similarities in species between mountain ranges within systems or regions) could have contributed to making the discussion more interesting. As such I think the paper would benefit from additional work including on the discussion before publication.

Below are some more detailed comments.

Introduction

The first part of the introduction is the somewhat "traditional" sales pitch about why mountains matter and how biologically rich they are. For this type of piece and for the topic of invasive species, I am not sure this is useful and necessary. I would recommend getting more effectively to the point of invasions in mountains and the threat those represent for standing biodiversity and NCPs. Accordingly, I would suggest adding a few more lines of content on the knowledge gained about plant invasions in mountains (around line 66).

I. 20: climate is not a threat, climate change is one

I. 21: do you really "close the gap"? I think you contribute to closing it. I would recommend revising the formulation

I. 24: you seem to assume that documentation is given by occurrence data. However, documentation can possibly also consist of other information sources. Accordingly, I would avoid the formulation "the most documented species" and opt to talk about the species for which most occurrence data exist.

I. 28: are they studied or merely observed?

I. 28: are there species occurring only in PAs?

I. 42 : where does this value come from?

I. 43: citing Koerner 2004 seems pertinent here

I. 44-46: I am a bit at odds with the notion that mountain "regions" play an "evolutionary role". Is the fact of being "a museum" an evolutionary role? Moreover the notion of mountain region is vague. I would avoid the term "region"

I. 53-55: I think this sentence requires a reference to make the claim that the portion of research is "significant" (whatever "significant" means).

I. 57: does ice cover retreat belong together with land-use change or rather with climate change?

I. 59: missing a dot at the end

I. 67: you opened a parenthesis but didn't close it.

I. 71: what do you mean with "known alien occurrences"

I. 72-73: one doesn't "study" a research question but one "addresses" or "answers" a research question

I. 73-74: the first "alien" is not needed

I. 73-77: format, either 1), 2), etc or 1., 2., etc

I. 74: do you really study the dynamics? And are we talking about spatial or temporal dynamics or both? Methods

I. 81-84: why is this sentence relevant and needed?

I. 81: the authors refer to an inventory that was not developed based on expert opinion as such, i.e. no expert was consulted in the process of polygon delineation.

I. 81: the authors mention previous inventories in plural. If there is only one, no plural is needed. To the best of my knowledge, there is no other inventory comparable to that by Koerner et al besides the new one the authors used.

- I. 85: I am not fully sure I understand what layer the authors used. Did you use the highest possible hierarchical level, i.e. the one that includes the child ranges only (i.e. no parent range)?
- I. 87-88: I am not fully convinced about this argument. Why does this level of detail in particular allows this description? Would another layer at lower resolution not have allowed this? If not, why?
- I. 90: many of the polygons at the highest level (i.e. the child ranges you seemed to have used if my understanding is correct) correspond to mountain ranges. As such it is not correct to say that level 3 is the mountain range level. Moreover, level 1 is not strictly the continental level either since e.g. North America, which is level 1 for a mountain range such as the Central Green Mountains, is not a continent. Accordingly, I would remove the second part of the sentence between categories and (level 1).
- I. 92-93: what does it mean for a mountain to be restricted to mid-or high elevations. What would typically be a mid-elevation mountain? Moreover, you seem to mix bioclimatic belts (with an expression such as “lower-elevation belts”) with elevation in terms of masl. This sentence needs to be clarified. Moreover, as such it is unclear why this information (min/max elevation and bioclimatic belts) matters for your work.
- I. 94: what do you mean with “lower and upper limits”?
- Supplementary figure 1: what is the point of panel C? What is the value of it compared to min, max, mean, median of all mountains? Moreover, why is there a “but in upper...” in the legend of panel C? (I don’t understand the “but”).
- I. 102: do you mean that you subset the DASC0 to retain only data on vertebrates? The formulation seems complicated.
- I. 107: one doesn’t typically perform a procedure.
- I. 112-113: from what dataset did you discard records? It needs to be the DASC0 one but the formulation is not entirely clear.
- I. 115-116: to what does “that represent our study unit” refer to? A reminder of what the study unit is seems to be irrelevant here as you have described your study unit (i.e. mountain polygons at the highest hierarchical level) earlier.
- I. 116-117: why do you account for mountain area?
- I. 121: In what figure do you show this?
- I. 122: what do you mean with “flows”
- I. 122-130: in this section you explain how you depict flows but in the next paragraph you compare flows quantitatively. How did you quantify the flows that you depict?
- I. 148-150: how and why does overlapping alien records over PA polygons enables extracting information on PA categories?
- I. 150: how did you estimate this and why can you only estimate and not calculate?
- I. 151-153: how did you calculate it?
- I. 163-165: from where did you extract the three variables? It needs to be made clearer in lines 160-162 that Table 1 lists data sources as this sentence is currently not clear enough.
- I. 167: what do you define as a “major” port?
- I. 176: how relevant is it to use a mean roughness value per mountain?
- I. 185-188: for clarity, the second part of the sentence (i.e. inverse...) would better belong in a separate sentence.

Results

Figure 1: what do the circles represent?

Supplementary Table 2: I question the usefulness of such a table as the mountain ranges that are listed are for most of them known by very few people. Accordingly, I would at least name the larger mountain systems of which they are part or display this information on a map.

I. 233-235: this sentence is confused and needs to be reformulated. Also please clarify what you mean with “compile”. Also what is the difference between records and occurrences. If there is none I would recommend using only one word.

I. 243: why “on” each major vertebrate group. Also whether those are major groups or not is irrelevant. Those are simply the groups you chose.

I. 256: Africa is not a realm.

Figure 2: Oceania seems to be missing from the map

I. 285-287: please cut the sentence in two before the “nevertheless”.

I. 289: why are those values only approximate?

I. 291: what do you mean with “accounting for the available area”? Based on your method section, you do not intersect the PAs with the mountain polygons. Accordingly, when you talk about area, you don’t consider the mountainous extent of the PA in your calculation but the total extent of the PA. Yet, in many cases, PAs are not entirely mountainous or non-mountainous and only fragments of PAs are in mountains. How do you account for this in your calculations?

Discussion

Patterns of distribution: I find this discussion quite shallow with some common places such as the lack of monitoring and regret that it does not offer more suggestions for the observed patterns. For example, what would explain the disproportionate number of alien bird species in the Cairngorms for instance? And how would the distribution of richness change if the UK were left out? Would we learn something more? Would we also understand certain of those results better if analyses were performed on subgroups? For instance, are many alien bird species in the UK from one subgroup and would there be an explanation for this? Are alien mammals mostly large mammals or micromammals? Whereas I recognize that such a paper is not about explaining each result in detail, reflecting on possible reasons for the observed patterns (policies, history, etc) would I believe inform the thinking we perform on alien species management and in fact also the design of monitoring programs. More monitoring is not the solution if the monitoring is not designed carefully and if the design is not informed by some hypotheses. Succinct suggestive explanations appear only in the next section on lines 402-407

I. 364-366: this is certainly right but would it not nevertheless be possible to nuance a bit this discussion point by elaborating on what could be expected based on the explanatory variables you have looked at? Could you make approximate predictions of what to expect given e.g. the road densities, etc? And could you refer to existing data on invasive plants to

refine a bit your argument?

Flows: In this section I am equally missing explanations and hypotheses for the observed patterns and occasional differences with previous results. On l. 380-384 for example, you point to differences between your results and previous ones. How do you explain those differences?

l. 378-380: this would better be two sentences.

l. 385: what do you mean with "class level" and with "major flows"?

l. 390: from my understanding, the previous sentences are about intra-continental flows whereas here, while referring to the previous content, you talk about internal flows, which seems to be the topic of the next sentence only.

Protected areas: in line with previous comments, this discussion does not offer much perspective. In the case of birds for instance, the type of PA is not a barrier for an alien bird from one mountain range to colonize another one. In fact one could argue that the lack of disturbance within a strict nature reserve might play in favour of an alien bird species establishing itself there. Interesting analyses could tackle the question of whether alien species are similar or identical between nearby mountains for instance.

l. 408-411: this would best consist of two sentences.

l. 411-412: is this observation relevant without a distinction between the fractions of PAs that are in mountains and not?

l. 408-415: this reads more as a repetition of the results than as a discussion of the results.

Drivers: This section of the discussion remains at a very superficial level from my point of view. How can you explain the links with the information you provide and the patterns you detect between mountain ranges? Moreover, the section ends with little perspective for alien species research and management in mountains. Is a section missing that would offer interesting outlooks?

l.452-453: this sentence is grammatically incorrect.

(Remarks on code availability)

Reviewer #3

(Remarks to the Author)

This manuscript is an impressive attempt to describe the presence of alien vertebrate species in the mountains of the world. The authors have assembled previously available data from alien species presence at the world's scale (DASCO database) and analyzed it with generalized linear mixed models and ten different potential drivers to try to unravel possible correlations between alien vertebrate richness (AVR) and these drivers. As far as I am aware this is the first attempt to provide an account of the large-scale pattern of AVR of mountain areas. The authors have made a very thorough effort in analyzing such huge dataset and produced a set of nice and colorful plots of the world's distribution of AVRs which is combined with the above-mentioned statistical analyses. The analyses have been performed for all species together or by animal groups (fishes, amphibians, birds, reptiles and mammals). Overall, the results are interesting and worth being published. The paper is very well written and generally clear although I just found some few minor issues (detailed below).

My main concerns with the present version of the paper are related with (1) the validation of the original GBIF data, that has data from many different sources, of which many are entered through Citizen Science programs, and it is not clear how has it been validated. The authors basically refer to the paper of Seebens & Kaplan (2022) for details. However, as far as I understand, the methodology of DASCO does not filter by types of data into GBIF. These data could actually be filtered by looking at the metadata, to at least, check possible mistakes in the identification of species. Citizen Science programs such as iNaturalist generate a huge amount of data that goes into GBIF, and may contain significant identification errors. I understand it might not be easy to filter out possible errors such as wrongly identified species, but this might be a source for errors, and it would be wise to at least provide an assessment of the validity of the data. This could go into the supplement, but considering this information is the source information for the paper, I think is really necessary. A second issue related with the use of the GBIF database is that most of the data are from some particular areas of the world, while the rest are extremely under-represented. This is something already mentioned in the manuscript, but I am not sure how much does this affect the statistical models. Somehow this should be taken into account in the analyses, not just mentioning it in the discussion. Maybe doing an analysis separated for the northern hemisphere and the rest could be an option to test for this bias in the data.

2) The use of the IUCN Red List to validate the, that is a list of threatened species that have undergone an extinction risk assessment, but does not contain many species, that are simply not threatened, that contain some of the most invasive species. So the authors claim they "double-checked the alien status of each occurrence by performing a spatially explicit filtering procedure using available geographic information on species' native ranges...". I suggest that you explain which proportion of the 717 species were not able to assess with IUCN polygons. I am not suggesting that the work is not correct, but that you should provide a fair assessment of the validity of the results presented. A second aspect that might also be relevant for relatively small mountain areas is that many distribution maps of IUCN are not very accurate at all. Usually they come with some altitudinal range of the species distribution, which could be taken into account to further filter the polygon. Did you take that into account?

3) The concept of mountains is well defined, and mountain ranges have been identified by previous studies throughout the world. But also, it is well established that there are different types of mountain ranges, since some of them are relatively low in altitude and others are higher, referred as "high mountains". These different types of mountains have very different characteristics that are very relevant for the natural distribution of species (e.g. in the Northern Hemisphere fish are not native from high altitude aquatic ecosystems, but they are native in the lower mountains). I was disappointed by seeing that the species distribution with elevation was left to what is Supplementary Figure 5, that basically is a density plot of richness. It would have been very interesting to see a more in-depth analyses comparing "high mountains" from the rest.

Minor comments:

Title: I think the title should be a bit less ambitious. I encourage the authors to include richness, so that people understand that it is the main goal of the paper. You assess the occurrence of the species (with many considerations, as described above) but by no means their spatial distribution. For example, you might have a very small AVR value, but if these species are invasive and have a wide distribution the mountain range in question will have a very bad conservation status.

Statistical analyses. Overall, the main GLMM analyses are clear, they provided a repository of the script. However, they have performed a lot of analyses which are not so clearly described. For example, the randomization of alien species composition (L134-141) how was it performed? What about the maps of Figure 1? Can you please double check that all statistical analyses used are clearly cited?

I wonder why did you use a linear method (Generalized Linear Mixed Model (GLMMs) for analyzing your data, that is basically not linear. Why did not use a Generalized additive model (GAM) (Pedersen, Miller, Simpson, & Ross, 2019; Wood, Goude, & Shaw, 2014) approach that is more robust and allows for modelling non-linear variables? Did you check for linearity of the data and transformed them? I understand the case of area, but what about the other variables?

Table 1. Sampling effort is referred to sampling completeness in the text. Can you use only one term throughout the manuscript?

L221. At this point it would be good to mention that the data is available and the link at the end of the paper. However, why not use a repository that provide a DOI? I think it is a better idea. The Github repository is more for sharing software than actual data. On the other hand, it would be interesting to have a table in the supplement with the list of the different species, so that one may easily check what species are in the dataset.

Another aspect is the differences in the number of species within each animal group. Some groups such as amphibians have much less species than the others. Is this because there are not many alien amphibians, or a result of a data bias (being underrepresented) or that there are fewer amphibian species and therefore this is expected? Is the number of alien species a result of the number of species available or there are differences on the invasiveness of the different animal groups?

L250. I think that the chord diagram is not really a network analyses.

L260. Replace Table 2 to Table 3.

Figure 2 and the rest of figures and Tables. You cite some papers not following the numbers that are used in the text, which makes it more difficult to find the reference in the reference list. I would suggest to include the number after the reference, even if you cite the author and the year.

L310-319. It seems unnecessary to cite estimate [\pm SE] every time. Just at the first case should be enough.

L444. There are no "taxon" specific models, I would say they are "animal group" model. One may think you did run a model for each taxon.

Supplementary Figure 5. Figure S5. Why to names? I wonder why you scale the plots up to 6000 meters while there are no species above 4000 m. I think it would be better to cut the plot at the last elevation.

References

- Pedersen, E. J., Miller, D. L., Simpson, G. L., & Ross, N. (2019). Hierarchical generalized additive models in ecology: an introduction with mgcv. *PeerJ*, 7, e6876.
- Seebens, H., & Kaplan, E. (2022). DASC0: A workflow to downscale alien species checklists using occurrence records and to re-allocate species distributions across realms. *NeoBiota*, 74, 75-91.
- Wood, S. N., Goude, Y., & Shaw, S. (2014). Generalized Additive Models for Large Data Sets. *Journal of the Royal Statistical Society Series C: Applied Statistics*, 64(1), 139-155. doi:10.1111/rssc.12068

(Remarks on code availability)

Version 1:

Reviewer comments:

Reviewer #1

(Remarks to the Author)

After careful reading the revised manuscript and the response letter, I am overall satisfied with the new version and the explanations. The manuscript has also been significantly improved. However, I think there are still two main aspects needs to be further strengthened.

First, although the authors argue that this work focuses on the mountain areas and comparison of patterns with other lands is beyond the scope of this study. I am sorry that I cannot completely agree with this opinion. It is necessary to establish a reference point to understand the situation of biological invasion in mountain areas. In fact, many alien species have also invaded in other regions far from mountains. Such comparison will be very necessary and important to provide insights into the pattern and influencing factors of alien species in mountain areas. For example, a recent study exploring the biological invasions on indigenous people's lands (IPLs) also compared the alien species richness on IPLs and other lands after accounting for sampling intensities.

Second, regarding the role of invasion meltdown in explaining the invasion outcomes, the authors argued that it may have data resolution and analysis circularity issues. Although the country level data is indeed less accurate, but it is the currently only available data and it is worth to do at least an additional analysis provided in supplementary materials. Regarding the variable circularity issue, I suggest that the authors can use the prior presence of other alien species, which would not require data from the vertebrate response variable.

(Remarks on code availability)

Reviewer #2

(Remarks to the Author)

I reiterate the introductory comments provided during the first round of reviews.

Research on alien species in mountains is largely biased towards plants. Accordingly, by looking at alien vertebrate species in mountains this paper contributes to filling a gap and is an important contribution to our understanding of the threat that invasive alien species represent for mountain ecosystems. As such I very much welcome this effort that I see as important. The authors make smart use of existing data on alien species and map them onto mountain ranges to detect spatial patterns in species richness, including in PAs. Additional analyses generate a number of results on the flow of species and on possible drivers.

One main comment on the first version of the manuscript was that the discussion was quite shallow. The authors have worked on it to enrich it a bit and I welcome this effort. Besides some smaller comments listed below I would like to note that although the discussion has become better, it remains somewhat unsatisfactory from my point of view. The additions do in fact read a bit as if they were added somewhat quickly, with little attention to the English and to the logic of some arguments. Moreover, from my point of view the discussion still lacks useful information on what do do with those results, on ways forward. It reports and clarifies in light of existing results in other taxonomic group but offers little if any enlightening thoughts or ideas on how to concretely make use of those results in research and management (besides in the context of monitoring). I will argue again for taking the time to think beyond observations in this discussion and to formulate ideas in sharp English.

l. 41. Statements about how much land surface mountains cover are useless, unless it is made clear that this is based on a given definition of mountains and why that definition is chosen. The 25% is based on the WCMC definition and a priori there is no good reason to apply that definition to work on mountain biodiversity and not that of e.g. Koerner et al, or Snethlage et al. The authors use the mountain inventory by Snethlage et al0., which in its original version is not based on the WCMC definition.

l. 54-56. Since ice cover retreat belongs to climate change (cf answers to the comment), it should be mentioned before land-use change, together with climate change.

l.69. "the cases of" can be removed to ease the reading and the sentence modified accordingly. There is a growing body of literature on fish invasions in mountain lakes to which you could refer in more general terms.

l. 92. In what does the hierarchical approach provide high accuracy? The rivers do indeed help achieve accuracy, but this is unrelated to the hierarchical approach.

l. 94. The version number is v2.0 and not v.20.

Comment to line 92-93 of the first manuscript: I don't seem to find the sentence you quote in your answer.

l. 114-117. This sentence is long and unclear. I recommend revisiting it.

l. 129-130. You mean you discarded from the vertebrate records' subset those records falling The edits are not very clear.

l.138. You mention Figure 2 before Figure 1 in your manuscript.

Figure 2. The centroids are hardly visible. This Figure needs to be revisited. Are the centroids really needed?

l. 175-178. This points was raised before and I am still at odds with the approach taken. Why did you not intersect your PA layer with the mountain polygons to get only the proportion of PAs that is truly in mountains? I also don't quite understand the argument you propose for using the full PA area when only a fraction is in mountains. I acknowledge the answer you

gave but still don't understand why you don't account for the mountainous fraction of PAs. Why would the full size of the PA and not the size of the mountainous fraction have an effect? Would the edge effect not apply equally if you considered the mountainous fraction?

l.202 (former line 176). the authors argue that rough terrains provide a wider suite of habitats, which I get but does a mean value for an entire mountain range make sense? The roughness could be high in some part of the mountain ranges and very limited elsewhere. For larger mountain ranges, a mean value seems of little relevance.

l.257: what do you mean with "considerable"? Is it more or less than expected, than some sort of mean? This is subjective.

l. 257-259. What is the value of such a statement if you don't say what mountains those are?

l. 259. Why use "still"?

l. 259-262. You talk about positive deviations (from what?) and residuals. What are those residuals, how are they calculated, and what do they represent?

l. 261-262. I don't quite understand this sentence: you are saying that only a few mountains in Mexico, China, Australia, and Brazil are among those with positive residuals above 10 but below 25. What or where are the "other mountains" (since you say "among those" it means that there are others).

l. 267. "on the contrary" and not "contrary"

l. 284-288. You refer to Olsen's ecoregions in this section although in your methods you introduce them only in view of reporting on the global flows of species.

Figure 5: why use a different colour for the taxonomic groups then in figure 4?

l. 462-464. please revise the English

l. 468-471. The use of "on the other hand" and of "also" is unjustified. Also "reduced" as compared to what?

l. 473-475. Would the human presence necessarily increase the detection probability? This argument seems somewhat lost there. Do you have any reference or context that you could add to enrich this statement?

l. 484-488. Please reformulate by cutting this sentence in two. It is too long.

l. 488-490. Please revise the English.

l. 491. What do you mean with "vertebrate group models"?

l. 493-494. This should be a sentence on its own.

l. 491-494. Do you have some references from other studies to make your point with an example?

l. 494. Please use something else then "concerningly" to start the sentence. And why do you need to use "still"?

l. 497-499. Please revise the English and specify what repercussions you are talking about

l. 501. What specifically do you mean with "consequences for conservation"? Such a statement is of little interest if you don't expand on it (cf my general comment on the discussion)

l. 510. The "play an important role in" is unnecessary

l. 512-515. Making the point that the probability of detection increases with sampling effort in undersampled regions is not offering a particularly interesting insight. Moreover, the link between this argument and the fact that diverse environmental conditions could offer more opportunities is unclear. Please revisit this argument.

(Remarks on code availability)

Reviewer #3

(Remarks to the Author)

I have reviewed the changes made in the paper and the responses given by the authors, and I am pleased to see they have satisfactorily responded to all my corrections.

(Remarks on code availability)

RESPONSES to REFEREES

Reviewer #1 (Remarks to the Author):

Mountain areas are conservation hotspot of native biodiversity. However, we still understand little on the non-native species invasions in mountain areas at the global scale. This study used global dataset to analyze the distribution, flow, pattern in protected areas, and the drivers' of alien vertebrate species in mountains worldwide. So, this is a novel topic in invasion ecology. I provide some concerns and suggestions to further strengthen this manuscript.

R/ Thanks for your very detailed, and overall positive review. Below we address each of the points you mentioned

Line 40-51: For the first paragraph of the Introduction section, in addition to the importance of mountains in biodiversity conservation and providing ecosystem services, please also address the environmental change challenges the mountains are facing in the current era of global change.

R/ Thanks for the suggestion. The environmental change challenges are mentioned in the next paragraph (please see from line 52 onwards). In the first one, we aim to stress the importance of mountains for biodiversity and humans as well as the fact that their legacy is threatened. This serves to stress from the very beginning the need for studies in mountains in the context of global change, but also as a transition to the next paragraph where we elaborate more on specific drivers of global change.

Line 64-69: Although it is true that it remains little investigated on large-scale patterns of animal invasions in mountains, please provide more details on the invasion cases of alien animals at the local and regional scales.

R/ Thanks for the suggestion. We have added to the text the specific examples already cited in both cases. Now the text reads: *“Some examples are the cases of the crayfish (*Procambarus clarkii*) in California mountains and the wild goats (*Oreamnos americanus*) in Yellowstone*

National Park or the trout species (e.g. Salmo trutta or Oncorhynchus mykiss) and the alpine marmots (Marmota marmota) introduced in the Pyrenees^{33–36}.” (Lines 69-73)

Line 74: Please clarify what does “dispersal dynamics” mean here? How to confirm the spread stage along the introduction-establishment-dispersal invasion process?

R/ Thanks for the observation. To avoid misunderstandings, we have reformulated this question to clarify that we restricted this analysis to describe flows from native ranges to recipient mountain regions at the realm level. The new text reads *“What are the direction and magnitude of flows of alien vertebrate species reported in mountains between their native and recipient realms?”* (Lines 81-82)

Line 75: How to determine the independent role of PAs in resisting or buffering (the authors wrote here) biological invasions by controlling for the potential effects of human activities?

R/ Based on our data we can say whether alien vertebrate richness and abundance in mountain PAs differ among protection categories. Therefore, we simplified the presentation of this question in the text, accordingly: *“What is the incidence of alien vertebrates in different types of protected areas in mountains?”* Lines (82-83)

Line 97-110: As mountains are usually located in some natural regions far from urban areas, there might be literatures reported the occurrence of established non-native species using non-English-languages, which are needed to be included in the data collection (e.g., Amano et al. 2021).

Ref. Amano, T., V. Berdejo-Espinola, A. P. Christie et al. 2021. Tapping into non-English-language science for the conservation of global biodiversity. PLoS Biology 19: e3001296.

R/ Thanks for pointing this out and for the suggested reference. Indeed, we carefully considered extending our record compilation to the literature. However, we decided to restrict it to DASCOS dataset occurrences which are derived from GBIF. Despite the known potential biases

associated with these occurrences, they are contributed by many data holders who are not English-speaking. This is mirrored by the fact that GBIF is working on relieving language barriers working in association with translators in 10 different languages (please see <https://www.gbif.org/translators> for further details).

Nevertheless, in the new version we include this point in the discussion. We explicitly state that there is a need to fill knowledge gaps in understudied regions and propose to target efforts to recover cases documented in the literature in languages other than English as one major step forward. The related text now reads as: *“Consequently, we stress the need for further efforts to reduce these gaps, which are especially disproportionate in tropical regions, as has been documented also for other taxonomic groups⁶⁸. For vertebrates, we found few to no records of alien amphibians in regions like the Andes and most African mountains, and the same for fishes and reptiles in the mountains of mainland Asia. Disentangling whether this mirrors a lack of threat in such regions or an underestimation due to low sampling requires more research. On the one hand, an increase in monitoring initiatives is necessary to document new records, but also a targeted effort to retrieve and mobilize published information from the non-English-language scientific literature is an important first step to take⁶⁹. ”* (Lines 402-410)

Line 97-121: My major concern on the non-native species data is that the authors did not validate the establishment status of each non-native species. Compared with only non-native species occurrence data, it should confirm that whether these non-native species have established feral populations in mountain areas.

R/ Thanks for your observation. While obtaining such mountain-specific data on establishment for each species is impossible because the data is not available at that resolution for many species, it is important to notice that the regional polygons employed by the DASCO workflow to filter occurrences correspond to areas where the species are known to be established. This certainly helps to reduce the amount of occurrences within these areas that might refer to non-established populations. We agree that some populations of the species reported here may not be established, but this does not guarantee that they will not successfully establish in the future. Under such circumstances, we consider it better to adopt a conservative approach and

keep all reliable presences, including those potentially representing non-reproducing populations. For this reason and also considering that records of species in the wild already represent an important stage in the invasion process, we opted to analyze the whole set of alien species with presence records in mountains.

To make this point clear we added in the text that *“It is important to acknowledge that GBIF, and consequently DASCO, do not explicitly discriminate between casual and established self-sustaining populations, and therefore they might overestimate the current distribution range of invaders. Nevertheless, the DASCO workflow considers polygons representing areas where the species are known to be established, which reduces the bias mentioned above. Following a precautionary principle, we analyzed DASCO’s records, assuming that an overestimation is preferable over an underestimation or an incomplete description of the spread of alien species where casual populations are excluded because the latter may also exert impacts on biodiversity and ecosystem services.”* Lines 110-118

Line 155: How about the invasion meltdown effect? It is needed to include the number of prior presence of other established non-native species (e.g., Redding et al. 2019).

Ref. Redding, D. W., A. L. Pigot, E. E. Dyer, Ç. H. Şekerciöğlü, S. Kark, and T. M. Blackburn. 2019. Location-level processes drive the establishment of alien bird populations worldwide. *Nature* 571:103-106.

R/ Unfortunately, such an analysis needs information on the temporal sequence of establishment of species and interactions among them which is not available or only exists for a subset of species at a much larger geographical resolution. Such is the case of the First Records database by Seebens et al. which does not allow for a specific focus on mountain ranges. In addition, we believe that such a predictor could introduce some circularity to our analysis because it would require data from the response variable, which is per mountain species richness.

Line 172: It is not clear that whether the topographic complexity has been incorporated into the climate change velocity (e.g., Sandel et al. 2011).

Ref. Sandel, B., L. Arge, B. Dalsgaard, et al. 2011. The influence of Late Quaternary climate-change velocity on species endemism. *Science* 334:660-664.

R/ Thanks for pointing this out. As mentioned in the text, topographic complexity was calculated as terrain roughness, a metric based on elevation information. In an independent step, we estimated climate change velocity solely based on temperature, as this metric, as proposed by Sandel et al. 2011, does not consider topographic complexity. Looking at the correlations among our predictors, these two variables and topographic complexity are only weakly correlated (Supplementary Figure 3), therefore we include the three of them in our analyses.

Line 197-198: For the species completeness variable, is it downloaded from an existing database or re-calculated in this study? Please clarify the data source information and provide the R code to generate this important variable.

R/ We calculated the species completeness variable using the data available in the supplementary material from Meyer et al. (2015) (see Table 1). This study provides csv. files for mammals, birds and amphibians at 1-degree resolution which contain information on inventory completeness. The inventory completeness calculates the difference between the expert estimation of species richness and the actual species richness recorded by GBIF. Using this data we created raster layers of completeness for each of the mentioned groups and then averaged them to create a final layer that represents a proxy of mean vertebrate sampling completeness.

To improve clarity, In the new version we added some additional details to the original description. Lines 221-225 read as: *"We used available data on inventory completeness for amphibians, birds, and mammals from⁵⁷. This variable is calculated as the difference between expert estimation of species richness and the actual species richness recorded by GBIF. We rasterized centroid estimations of inventory completeness for each taxonomic group and then averaged them to obtain a map with the global distribution of mean sampling completeness."*

Line 207: How about the rationale of the correlation value < 0.75 ?

R/ Thanks for the suggestion. We reformulated this sentence and added a reference supporting the threshold used to identify multicollinearity among our predictors:

Dormann, C. F., Elith, J., Bacher, S., Buchmann, C., Carl, G., Carré, G., ... & Lautenbach, S. (2013). Collinearity: a review of methods to deal with it and a simulation study evaluating their performance. *Ecography*, 36(1), 27-46.

Moreover, we clarify in the text that according to this reference, collinearity starts to severely affect model estimates when the correlation value is >0.7 .

Line 209: It is better to use an AIC-based model averaging method to calculate the relative *importance of each predictor variable*.

R/ We understand the suggestion. However, given that we scaled the predictors used in each model, we can also assess the relative importance of the variables based on the predictor coefficients. This has been done similarly in other studies evaluating correlates of invasion (e.g., Dawson et al. 2017). Moreover, since AIC-based evaluations require the implementation of multiple models accounting for different predictor combinations and the correct calculation of AIC values -especially for mixed models- is still a matter of debate among statisticians (e.g., Lee et al. 2023), we prefer to opt for the more simple and traditional approach of scaling which also improve interpretability as suggested by Schielzeth (2010).

Lee, Y., Rojas-Perilla, N., Runge, M. et al. Variable selection using conditional AIC for linear mixed models with data-driven transformations. *Stat Comput* 33, 27 (2023).

<https://doi.org/10.1007/s11222-022-10198-9>

Schielzeth, H. (2010), Simple means to improve the interpretability of regression coefficients. *Methods in Ecology and Evolution*, 1: 103-113. <https://doi.org/10.1111/j.2041-210X.2010.00012.x>

Line 210: It is not clear how the 'diamond shape' was quantified here.

R/ All the information on the mountain categorization protocol is available above in the manuscript (Lines 203-217). For the specific case of diamond mountains, we mention:

"Mountain ranges with skewness values between -0.5 and 0.5 were assigned to the diamond category." (Lines 212-213).

Line 212: In addition to the mountain systems nested in regions and in continents, it should also include the taxonomic identity of different taxa used in the analysis to account for the taxonomic sample non-independence.

R/ In the models we used alien species richness per mountain as the response variable. Since our study units are the mountains, this variable represents the total number of alien species occurring on each mountain ensemble. For this reason, including the species taxonomic identity is not feasible. Nevertheless, we considered this and used another approach which is the deconstruction of the overall pattern into five additional models, one for each taxonomic class (i.e. fishes, amphibians, reptiles, birds, and mammals). This allowed us to assess whether the significant drivers of alien richness in mountains found for the full dataset are consistent among taxonomic groups, accounting for the non-independence associated with the species taxonomic identity.

Line 221-249: As the authors have been able to account for the sample bias issue as mentioned in the Method section, it should be reported the bias-controlled pattern of distribution of established non-native vertebrates in global mountains.

We appreciate the suggestion. Accordingly, we have now created an additional figure showing the bias-controlled pattern. For this, we used the residuals of the regression between Alien Richness and the mean sampling completeness recorded for each mountain. A figure showing a map with the bias controlled pattern of distribution of alien richness and the regression plot is now available as Supplementary Figure 5. This figure is mentioned in the results text in lines 257-262, where we stated that *"Accounting for sampling completeness, additional mountains*

emerged as having higher than expected alien richness based on the sampling effort there conducted. Still, the mountains with the highest positive deviations are in Europe and the US with residuals ranging between 25 to 70 (Supplementary Figure 5). Few mountains in Mexico, China, Australia and Brazil are among those with positive residuals above 10 but below 25. ”

Additionally, does the observed non-native species altitudinal pattern reflect the classic mid-elevation peak pattern (i.e., mid-domain effect) of native species (e.g., Quintero and Jetz 2018)?

Ref. Quintero, I., and W. Jetz. 2018. Global elevational diversity and diversification of birds. *Nature* 555:246-250.

R/ This is an interesting observation, however, in our opinion, making such an isolated comparison would be biased, and conducting comparisons of elevational richness patterns between natives and aliens would certainly deserve a whole study. Moreover, our decision to restrict the discussion on within-mountain elevation patterns relies mainly on the fact that spatial uncertainties of GBIF-derived data could be especially problematic in mountain regions. In such settings, large elevation differences can arise even in small horizontal distances, which may distort the actual elevation richness pattern. This would limit proper comparisons with existing literature such as the suggested paper by Quintero et. al., which did not use GBIF data. Regarding elevation, we show the analyzed records' altitudinal distribution pattern in the supplementary Figure 8 of the current version. Nevertheless, due to the potential inaccuracies above mentioned we avoided drawing strong conclusions from this particular analysis.

Line 309-328: How about the relative importance of each predictor variable in explaining the established non-native vertebrate richness in global mountains?

R/ Thanks for the suggestion. We focus our analysis on the statistical significance of predictors, their relationship with the response (positive or negative), and the effect size of each. These are the standard outcomes assessed in regression analyses, as implemented in our study. We understand that assessing the relative importance of predictors could complement the

assessment to some extent. However, this would require a step-wise approach, such as the one based on AIC, as suggested. Nevertheless, as we mentioned above and given the drawbacks related to the calculation of such values in mixed models (see Lee et al. 2023, above), we prefer our approach based on established procedures.

Line 350: In addition to these similarities with the global picture of non-native vertebrates, are there any differences observed in the mountain areas compared with other areas from the present study?

R/ We studied exclusively mountain areas; therefore, the suggested comparison is beyond the scope of our study.

Line 359: So, it would be interesting to see a sampling bias-controlled global pattern of established non-native vertebrates.

R/ Correct. As mentioned above we are now providing such a map.

Line 443-452: After the initial introductions by humans, the subsequent spread of the established non-native species may also be highly related with the natural dispersal abilities among taxa, which are unfortunately lacking here.

R/ Yes, we fully agree on this point. Although we did not consider the dispersal abilities of our species for data availability limitations, in the new version we acknowledge this drawback in the discussion. The new text reads: *“Deconstructing the general picture into vertebrate group models we found that driver effects vary across groups. This variation may be influenced by species intrinsic traits, such as the different natural dispersal abilities after initial introductions; unfortunately, we could not consider this trait due to the limited data availability. Concerningly, we still may expect that overall, most groups will continue spreading rapidly in the upcoming years, as recent evidence shows that the secondary spread of alien species is much faster than the spread of native ones and the velocity of climate change⁸²” (Lines 488-494).*

Finally, as mountains have rich endemic species just as what the authors have mentioned in the Introduction section, the authors need strengthen the Discussion by addressing the potential impacts of the established non-native vertebrate on mountain endemic species.

R/ Following your observation in this new version we mention the suggested idea in the discussion as: *“Concerningly, we still may expect that overall, most groups will continue spreading rapidly in the upcoming years, as recent evidence shows that the secondary spread of alien species is much faster than the spread of native ones and the velocity of climate change⁸². Such differences may lead to the replacement of native species by alien ones likely having strong repercussions in vulnerable regions such as mountain ranges. Since these regions are characterized by high endemism rates and many native species have very restricted distribution ranges⁸³, such turnover will likely lead to extinctions and biotic homogenization with important consequences for conservation as demonstrated for other taxonomic groups⁸⁴. Lines 491-499*

Reviewer #2 (Remarks to the Author):

Research on alien species in mountains is largely biased towards plants. Accordingly, by looking at alien vertebrate species in mountains this paper contributes to filling a gap and is an important contribution to our understanding of the threat that invasive alien species represent for mountain ecosystems. As such I very much welcome this effort that I see as important.

The authors make smart use of existing data on alien species and map them onto mountain ranges to detect spatial patterns in species richness, including in PAs. Additional analyses generate a number of results on the flow of species and on possible drivers.

Besides individual and relatively small comments to the introduction, methods, and result (yet some additional clarifications are needed in the method section) one of my main concern with this paper is that the discussion is from my point of view very/too shallow as it offers very few explanations for the obtained results and hardly informs alien species research, management, and policy in mountains. Accordingly, whereas the work per se is certainly useful and necessary, the significance is not fully achieved as I remain with the question of "and so what", "where do we go from there". I believe that additional analyses (e.g. by species subgroups, or that look at similarities in species between mountain ranges within systems or regions) could have contributed to making the discussion more interesting. As such I think the paper would benefit

from additional work including on the discussion before publication.

R/ Thanks for your positive feedback and constructive criticism. We followed your suggestion to expand the discussion and, accordingly, we now address the highlighted aspects in more detail. In the responses to your comments and the other reviewers' suggestions, we provide detailed explanations of such additions.

Below are some more detailed comments.

Introduction

The first part of the introduction is the somewhat "traditional" sales pitch about why mountains matter and how biologically rich they are. For this type of piece and for the topic of invasive species, I am not sure this is useful and necessary. I would recommend getting more effectively to the point of invasions in mountains and the threat those represent for standing biodiversity and NCPs. Accordingly, I would suggest adding a few more lines of content on the knowledge gained about plant invasions in mountains (around line 66).

R/ We appreciate the suggestion. We consider it important to keep this paragraph as it highlights the general relevance of mountains for biodiversity in a wider context and justifies the urgent need to document the pressures that such regions are experiencing. Also, our concern is that starting directly with the impacts of alien species on biodiversity and NCPs could be interpreted as overselling our study as it does not focus on impact quantification.

On the other hand, following your suggestion, we now highlight the existing knowledge on invasive species in mountains and provide a reference that summarizes it. The modified text in the new version now reads as: *"While studies on alien plants in mountains have already provided evidence of the vulnerability of these regions to alien species (see ³⁵ for a detailed review), it is necessary to assess other taxonomic groups to fully understand the threat posed by biological invasions in these regions."* Lines 73-76

I. 20: climate is not a threat, climate change is one

R/ Thanks for pointing this out. We have changed this to "climate change"..

I. 21: do you really “close the gap”? I think you contribute to closing it. I would recommend revising the formulation

R/ Thanks, we have included the suggested modification. It now reads (line 22): *“To contribute to closing this gap,...”*

I. 24: you seem to assume that documentation is given by occurrence data. However, documentation can possibly also consist of other information sources. Accordingly, I would avoid the formulation “the most documented species” and opt to talk about the species for which most occurrence data exist.

R/ Changed accordingly. Please see in lines 25 and 26.

I. 28: are they studied or merely observed?

R/ Here we meant “studied by us” in this work. We have now reformulated the sentence for more clarity. It now reads (line 29): “Almost 50% of the alien species included in our study”.

I. 28: are there species occurring only in PAs?

R/ This is something we had not considered but now that we checked the data we realized that this is the case for only ten species. We added this information to the results.

I. 42 : where does this value come from?

R/ The value comes from the reference cited at the end of that same sentence. We have now moved the citation closer to the value and, following the next suggestion, we cited Koerner 2004 when referring to the physical and climatic heterogeneity of mountains.

I. 43: citing Koerner 2004 seems pertinent here

R/ Done accordingly.

I. 44-46: I am a bit at odds with the notion that mountain “regions” play an “evolutionary role”. Is the fact of being “a museum” an evolutionary role? Moreover the notion of mountain region is vague. I would avoid the term “region”

R/ The species accumulation at a given location is an important factor in shaping species richness patterns, and it usually occurs over long temporal scales. Such accumulation depends on species persistence through time implying low extinction rates, and extinction is (along with speciation and dispersal) a major evolutionary process. Therefore, we consider that mountains serving as refugia that allow such species persistence (i.e. museums) play an evolutionary role. Indeed, this metaphor of biodiversity cradles and museums has been largely applied to describe the governing evolutionary processes behind the ensemble of strikingly diverse biotas in some areas of the world, with cradles having speciation rates higher than expected and museums having reduced extinction rates.

Nevertheless, if after this justification you consider this should be changed we are open to make the modification. In agreement with your other suggestion, we have removed the word region here and throughout the text where it was accompanied by the word mountain.

I. 53-55: I think this sentence requires a reference to make the claim that the portion of research is “significant” (whatever “significant” means).

R/ Thanks for the observation, we decided to reformulate the sentence to avoid confusion. Please see in current line 53.

I. 57: does ice cover retreat belong together with land-use change or rather with climate change?

R/ With climate change.

I. 59: missing a dot at the end

R/ Corrected.

I. 67: you opened a parenthesis but didn't close it.

R/ Corrected.

I. 71: what do you mean with "known alien occurrences"

R/ Here, we refer to occurrences reported and georeferenced, but we agree that this wording was not fully clear. We changed it to *"georeferenced alien records"*.

I. 72-73: one doesn't "study" a research question but one "addresses" or "answers" a research question

R/ Rephrased accordingly (please see line 79).

I. 73-74: the first "alien" is not needed

R/ Removed.

I. 73-77: format, either 1), 2), etc or 1., 2., etc

R/ Corrected.

I. 74: do you really study the dynamics? And are we talking about spatial or temporal dynamics or both? Methods

R/ What we study is the flow of vertebrate species from their native to recipient realms. We have now reformulated the question accordingly: *"What are the direction and magnitude of*

flows of alien vertebrate species reported in mountains between their native and recipient realms?” Lines 81-82.

I. 81-84: why is this sentence relevant and needed?

R/ We consider this sentence as important to provide the readers with the context of the mountain inventory used because such a dataset is central for the analyses included in our study.

I. 81: the authors refer to an inventory that was not developed based on expert opinion as such, i.e. no expert was consulted in the process of polygon delineation.

R/ Thanks, you are right. We have reformulated the sentence: *“Compared to previous definitions of mountains that relied on expert opinion (e.g., ³⁹), this inventory is built based on...”*

Lines 90-91

I. 81: the authors mention previous inventories in plural. If there is only one, no plural is needed. To the best of my knowledge, there is no other inventory comparable to that by Koerner et al besides the new one the authors used.

R/ Thanks for spotting this. We have now changed the writing as shown in the previous comment.

I. 85: I am not fully sure I understand what layer the authors used. Did you use the highest possible hierarchical level, i.e. the one that includes the child ranges only (i.e. no parent range)?

R/ Yes, we used the layer containing the child ranges only. To make this point clear, we re-wrote this statement: *“...we only considered non-overlapping mountain polygons classified as “Mountains with well-recognized names” at the most basic mapping unit (i.e. mountains without smaller subdivisions) as our study units (n=4953 mountains across the globe)”*. Lines 94-96.

I. 87-88: I am not fully convinced about this argument. Why does this level of detail in particular allows this description? Would another layer at lower resolution not have allowed this? If not, why?

R/ Our point here is that using this hierarchical level we were able to describe in more detail the distribution of alien species and the characteristics of mountains at a finer scale than if we were considering clustered ranges conforming parent ranges. We changed the wording for clarification. Now the sentence reads: *“This level of classification allows us to describe in more detail the alien distributions and richness patterns in the mountains of the world and analyze their underlying drivers.”* Lines 96-98

I. 90: many of the polygons at the highest level (i.e. the child ranges you seemed to have used if my understanding is correct) correspond to mountain ranges. As such it is not correct to say that level 3 is the mountain range level. Moreover, level 1 is not strictly the continental level either since e.g. North America, which is level 1 for a mountain range such as the Central Green Mountains, is not a continent. Accordingly, I would remove the second part of the sentence between categories and (level 1).

R/ Done.

I. 92-93: what does it mean for a mountain to be restricted to mid-or high elevations. What would typically be a mid-elevation mountain? Moreover, you seem to mix bioclimatic belts (with an expression such as “lower-elevation belts”) with elevation in terms of masl. This sentence needs to be clarified. Moreover, as such it is unclear why this information (min/max elevation and bioclimatic belts) matters for your work.

R/ We included this information to highlight the heterogeneity of mountain types assessed. The definition of mountains based on terrain roughness allows the delimitation of mountains with different lowest elevations, which may have implications for example when estimating distances to ports, one of the variables evaluated here. The combination of minimum and maximum

elevations is also instrumental in calculating the elevation ranges covered by each mountain, which we hypothesize could influence richness as wider ranges would provide larger climatic gradients for species to occur. To avoid confusion we have rephrased the sentence which now reads: *“Note that mountains are defined based on the landscape roughness, and not by elevation, therefore many may also include adjacent lowlands when they are rough enough”* Lines 101-102.

I. 94: what do you mean with “lower and upper limits”?

Supplementary figure 1: what is the point of panel C? What is the value of it compared to min, max, mean, median of all mountains? Moreover, why is there a “but in upper...” in the legend of panel C? (I don’t understand the “but”).

R/ We meant lower and upper elevation limits. We have corrected it in the new version.

Regarding the Supp. Figure 1, the correct word in the caption is “both”, not “but”. This figure shows the heterogeneity of mountain types in terms of the elevation ranges they cover and supports the statement provided in the text stressing that mountains may include lowlands but in other cases, they have lower elevation limits at higher elevations.

I. 102: do you mean that you subset the DASC0 to retain only data on vertebrates? The formulation seems complicated.

R/ Thanks for the observation. For more clarity, we rewrote the sentence: *“We first subset DASC0’s data to retain only records belonging to vertebrate species.”* Line 119.

I. 107: one doesn’t typically perform a procedure.

R/ Change made. Lines 123-124 now read as : *“...we double-checked the alien status of each record by using a spatially explicit filtering procedure...”*.

I. 112-113: from what dataset did you discard records? It needs to be the DASC0 one but the formulation is not entirely clear.

R/ That is correct, from the DASCO data. We have now written this explicitly."... *from the vertebrate occurrences subset from DASCO obtained before*". Lines 130-131.

I. 115-116: to what does "that represent our study unit" refer to? A reminder of what the study unit is seems to be irrelevant here as you have described your study unit (i.e. mountain polygons at the highest hierarchical level) earlier.

R/ Removed.

I. 116-117: why do you account for mountain area?

R/ Given the well-known species-area relationships in ecology, accounting for the differences in the areas of the studied regions is a common procedure in macroecological approaches that allow for more balanced comparisons among the study units.

I. 121: In what figure do you show this?

R/ In Figure 1. We have added to the text the respective reference.

I. 122: what do you mean with "flows"

R/ We have clarified this in the text as: "*(i.e., species displacements from native to alien regions)*." Lines 140-141.

I. 122-130: in this section you explain how you depict flows but in the next paragraph you compare flows quantitatively. How did you quantify the flows that you depict?

R/ We have added a new sentence explaining this. Lines 146-149 now read as: "*...Similarly, for each alien occurrence in our dataset, we assign the realm in which they occur by overlapping the realms and the alien occurrence layers. Using this information, we quantified the number of species moving from each native realm to the alien ones for each taxonomic group. We then used this input to depict the flows between native and invaded realms for ...*"

I. 148-150: how and why does overlapping alien records over PA polygons enables extracting information on PA categories?

R/ Because each polygon contains associated information that includes PA categories. We included “available from the WDPA layer” to make this clearer. (See line 168)

I. 150: how did you estimate this and why can you only estimate and not calculate?

R/ Word changed.

I. 151-153: how did you calculate it?

R/ To clarify this, we added the respective information as follows: “...by summing up the taxa with occurrences documented within PAs”. Lines 172-173.

I. 163-165: from where did you extract the three variables? It needs to be made clearer in lines 160-162 that Table 1 lists data sources as this sentence is currently not clear enough.

R/ It is now mentioned in the previous sentence (line 188) that Table 1 contains all data sources used.

I. 167: what do you define as a “major” port?

R/ We removed the word “major” to avoid confusions.

I. 176: how relevant is it to use a mean roughness value per mountain?

R/ We hypothesized that rough terrains provide a wider suite of habitats that could be occupied by a larger number of alien species. The relevance of the metric will be tested at the light of this hypothesis.

I. 185-188: for clarity, the second part of the sentence (i.e. inverse...) would better belong in a separate sentence.

R/ Thanks, suggestion implemented.

Results

Figure 1: what do the circles represent?

R/ The circles represent the location of the centroids of the mountains studied. We have added this information to the caption.

Supplementary Table 2: I question the usefulness of such a table as the mountain ranges that are listed are for most of them known by very few people. Accordingly, I would at least name the larger mountain systems of which they are part or display this information on a map.

R/ Thanks, we agree with both suggestions. Therefore, although we are keeping Supplementary Table 2, now we are showing on a map examples of mountains in this top ranking (please see the new Figure 1). This certainly provides a better geographic context, and in addition to the specific mountain name, we detailed the name of the larger mountain system each mountain belongs to and the number of species recorded for each specific mountain.

I. 233-235: this sentence is confused and needs to be reformulated. Also please clarify what you mean with “compile”. Also what is the difference between records and occurrences. If there is none I would recommend using only one word.

R/ Thanks for the suggestions. In this new version, we stick to the term “records” throughout the manuscript.

I. 243: why “on” each major vertebrate group. Also whether those are major groups or not is irrelevant. Those are simply the groups you chose.

R/ Thanks for the suggestion. Change made.

I. 256: Africa is not a realm.

R/ Corrected.

Figure 2: Oceania seems to be missing from the map

R/ Thanks for pointing this out, given the small area of such territories and the limited size of the map Oceania was indeed not visible. In this new version we opted to add ellipses to indicate the regions corresponding to this biogeographic realm.

I. 285-287: please cut the sentence in two before the “nevertheless”.

R/ Done.

I. 289: why are those values only approximate?

R/ We provided the exact values in this new version.

I. 291: what do you mean with “accounting for the available area”? Based on your method section, you do not intersect the PAs with the mountain polygons. Accordingly, when you talk about area, you don’t consider the mountainous extent of the PA in your calculation but the total extent of the PA. Yet, in many cases, PAs are not entirely mountainous or non-mountainous and only fragments of PAs are in mountains. How do you account for this in your calculations?

R/ Yes, this is correct. We accounted for the area considering the full surface of each PA. In our opinion, the full size of the PA is relevant as it may have implications on invasion dynamics. For example, invasion risk could be affected by edge effects, which in turn may differ between PAs of different sizes. Indeed, it has been shown that the total surface area of the PA is strongly and positively correlated with species richness across terrestrial vertebrates and many invertebrate groups (Liu et al. 2022). For this reason, we would prefer to keep the analysis as implemented.

To improve clarity, we added the above explanation to the Methods section. The respective text reads as: *“This procedure was then repeated accounting for the surface area of the studied PA’s. Despite some PAs may have portions of their surface outside mountains, we considered the full surface as PA size has been shown to be a strong predictor of alien richness for many animal groups, including terrestrial vertebrates and several invertebrate groups”* Lines 174-175

Liu, X., Blackburn, T.M., Song, T. et al. Animal invaders threaten protected areas worldwide. *Nat Commun* 11, 2892 (2020). <https://doi.org/10.1038/s41467-020-16719-2>

Discussion

Patterns of distribution: I find this discussion quite shallow with some common places such as the lack of monitoring and regret that it does not offer more suggestions for the observed patterns. For example, what would explain the disproportionate number of alien bird species in the Cairngorms for instance? And how would the distribution of richness change if the UK were left out? Would we learn something more? Would we also understand certain of those results better if analyses were performed on subgroups? For instance, are many alien bird species in the UK from one subgroup and would there be an explanation for this? Are alien mammals mostly large mammals or micromammals? Whereas I recognize that such a paper is not about explaining each result in detail, reflecting on possible reasons for the observed patterns (policies, history, etc) would I believe inform the thinking we perform on alien species management and in fact also the design of monitoring programs. More monitoring is not the solution if the monitoring is not designed carefully and if the design is not informed by some hypotheses. Succinct suggestive explanations appear only in the next section on lines 402-407

R/ Perhaps the perception of a shallow discussion is partially explained by the way we structured this section. In terms of structure, we approached each of the subsections in the same order as in the methods and results. Nevertheless, we now go into more detail about the suggestions for the observed patterns in the subsection of the discussion related to the drivers tested. As explained in Table 1, all drivers are related to a specific hypothesis. While your suggestion of comparing regions, species taxonomic or trait-based subgroups is certainly interesting, the nature of the data does not allow such comparisons. This is beyond the scope

of our study which is testing the specific drivers presented in the introduction. One exercise we did as sensitivity analyses for identification of relevant drivers was running all models excluding 1. the outliers in the response variable (i.e. mountains with >60 alien species), 2. the upper quartile of the distribution of alien richness in mountains and 3. the upper and lower quartile of this same distribution. The results were highly consistent with those shown in the main text. In this new version, we provide a new figure (see Supp. Fig 7) in the supplementary material showing these results. Moreover, we explicitly mentioned this in the text as: “Sensitivity analyses fitting models with different subsets of mountains showed consistent results overall (Supplementary Figure 7).” Lines 367-368.

I. 364-366: this is certainly right but would it not nevertheless be possible to nuance a bit this discussion point by elaborating on what could be expected based on the explanatory variables you have looked at? Could you make approximate predictions of what to expect given e.g. the road densities, etc? And could you refer to existing data on invasive plants to refine a bit your argument?

R/ Thanks for the useful observation. Indeed, we touch more on this point later in the discussion, in the section “Drivers of alien richness in mountains”. Following your suggestion we added text on existing evidence for plants to strengthen the argument. The new text reads as *“These results suggest that independent of the biological particularities of each group, the emerging richness patterns of alien vertebrates in mountains are strongly tied to infrastructures that promote connectivity and facilitate the human-driven spread of introduced species, but also to sampling efforts, and heterogeneous landscapes. Supporting this, previous monitoring efforts following standardized protocols (e.g., ³²) have revealed that -at least for plants ⁸⁴- roads play an important role in providing favorable habitat and anthropogenic dispersal routes for many alien species in mountains (reviewed in ⁸⁵). Moreover, our findings indicate that as sampling efforts improve in understudied regions, the detection of alien vertebrates may increase, particularly in areas with rugged terrain, where the diverse environmental conditions could offer more opportunities for introduced species to establish.”*

” Lines 501-511.

Flows: In this section I am equally missing explanations and hypotheses for the observed patterns and occasional differences with previous results. On l. 380-384 for example, you point to differences between your results and previous ones. How do you explain those differences?

R/ Given the complexity and number of total flows it is hard to delve into all the observed patterns, but in this new version we add some thoughts in this section. For example, regarding the point mentioned, we hypothesized that the differences in this specific case rely on the fact that the cited reference is about plants and our study is on vertebrates which have percentages of deliberately introduced species, for example as pets, probably higher. This may be a reason for the reverse flow from the south to the richer (and hence more pet-keeping) north. Lines 428-432 now read as: *“Variable intensity in the flows between the southern and northern hemispheres could be partially determined by specific introduction pathways that are imposed by activities such as the pet market, which is in turn influenced by cultural differences in pet-keeping traditions across regions”⁷⁴.*

l. 378-380: this would better be two sentences.

R/ Done

l. 385: what do you mean with “class level” and with “major flows”?

R/ Class level refers to each of the taxonomic classes studied (i.e. Actinopterygii, Amphibia, Reptilia, Aves and Mammalia) and major flows refer to the flows involving the largest number of species. We believe that the following sentence (line 434) provides enough context and makes clear what we are referring to.

l. 390: from my understanding, the previous sentences are about intra-continental flows whereas here, while referring to the previous content, you talk about internal flows, which seems to be the topic of the next sentence only.

R/ Perhaps the confusion here is that our study units are realms while in some of the cited

studies they are continents. For that reason, we used “within-realms”, “intra-continental” and “internal” flows.

Protected areas: in line with previous comments, this discussion does not offer much perspective. In the case of birds for instance, the type of PA is not a barrier for an alien bird from one mountain range to colonize another one. In fact one could argue that the lack of disturbance within a strict nature reserve might play in favour of an alien bird species establishing itself there. Interesting analyses could tackle the question of whether alien species are similar or identical between nearby mountains for instance.

R/ This is a great point, unfortunately it goes beyond the scope of our study. This would require independent analyses comparing the similarity between pairs of mountains accounting for the pairwise distances. From there one could be able to break such relationships in the studied taxonomic groups and evaluate the differences depending on the mobility of each group. Definitely interesting enough to deserve a whole study.

I. 408-411: this would best consist of two sentences.

R/ Done

I. 411-412: is this observation relevant without a distinction between the fractions of PAs that are in mountains and not?

R/ Yes, it is relevant because we used only records falling within mountains. This means that the absolute value provided here is not overestimated as we restricted the count to the records occurring within the mountain portion of each PA.

I. 408-415: this reads more as a repetition of the results than as a discussion of the results.

R/ In this paragraph, the first two sentences refer to another study; therefore, they are not a repetition of our results. In the next sentences we compared such existing evidence with our

findings and in doing so it is necessary to summarize and highlight our results. To make this comparison more explicit, we have now added the percentage of the total number of PAs that 820 PAs represent. Regarding the sentence on the dominant taxonomic groups we clarify that this is expected due to the composition of our dataset. We do not delve into more details because this was already discussed in the section on “Patterns of distribution of alien vertebrates in mountains worldwide”.

Drivers: This section of the discussion remains at a very superficial level from my point of view. How can you explain the links with the information you provide and the patterns you detect between mountain ranges? Moreover, the section ends with little perspective for alien species research and management in mountains. Is a section missing that would offer interesting outlooks?

R/ In this new version we have delved more into the discussion of our results. The modifications on this section also respond to observations made by the other reviewers. Please see lines 491-502;509-515 and 517-526.

I.452-453: this sentence is grammatically incorrect.

R/ We reformulated this sentence as follows: *“For example, the local increase in minimum temperatures during the last century was a significant and positive predictor for ectotherms (i.e. fishes, amphibians, and reptiles).”* Lines 516-518. Moreover, we extended the discussion of this interesting finding adding the following: *“This suggests that rising minimum temperatures due to anthropogenic emissions could enable the expansion of these groups into higher elevations, which is particularly concerning considering that the climate in mountains is expected to change three times faster than the global average 88. Evidence of altitudinal range shifts due to climate change has been documented in various taxonomic groups within their native ranges 89. However, our findings suggest that upslope shifts could also be possible for ectotherms in non-native regions. In these cases, human-facilitated introductions could substantially contribute to the spread of species that otherwise would be limited by their relatively poor dispersal abilities compared to birds and mammals.”* Lines 516-526.

Reviewer #3 (Remarks to the Author):

This manuscript is an impressive attempt to describe the presence of alien vertebrate species in the mountains of the world. The authors have assembled previously available data from alien species presence at the world's scale (DASCO database) and analyzed it with generalized linear mixed models and ten different potential drivers to try to unravel possible correlations between alien vertebrate richness (AVR) and these drivers. As far as I am aware this is the first attempt to provide an account of the large-scale pattern of AVR of mountain areas. The authors have made a very thorough effort in analyzing such huge dataset and produced a set of nice and colorful plots of the world's distribution of AVRs which is combined with the above-mentioned statistical analyses. The analyses have been performed for all species together or by animal groups (fishes, amphibians, birds, reptiles and mammals). Overall, the results are interesting and worth being published. The paper is very well written and generally clear although I just found some few minor issues (detailed below).

R/ Thanks a lot for the very positive evaluation of our manuscript.

My main concerns with the present version of the paper are related with (1) the validation of the original GBIF data, that has data from many different sources, of which many are entered through Citizen Science programs, and it is not clear how has it been validated. The authors basically refer to the paper of Seebens & Kaplan (2022) for details. However, as far as I understand, the methodology of DASCO does not filter by types of data into GBIF. These data could actually be filtered by looking at the metadata, to at least, check possible mistakes in the identification of species. Citizen Science programs such as iNaturalist generate a huge amount of data that goes into GBIF, and may contain significant identification errors. I understand it might not be easy to filter out possible errors such as wrongly identified species, but this might be a source for errors, and it would be wise to at least provide an assessment of the validity of the data. This could go into the supplement, but considering this information is the source information for the paper, I think is necessary.

R/ Regarding this concern, it is important to note that only research grade observations from iNaturalist that meet specific standards enter into GBIF. These requirements include an

agreement on the species identification by the involved community. More details on the methodology and quality control applied to classify such data can be accessed here:

<https://www.gbif.org/dataset/50c9509d-22c7-4a22-a47d-8c48425ef4a7>

We consider that this error check mechanism is likely more efficient than anything we could do post-hoc. On the other hand, even if present, misidentifications are likely random in space and environment, they hence add noise but not bias. It can result in precise numbers getting a broader confidence interval, but statistical tests based on these numbers have rather higher Type II than Type I errors, therefore the effects found here should be robust.

A second issue related with the use of the GBIF database is that most of the data are from some particular areas of the world, while the rest are extremely under-represented. This is something already mentioned in the manuscript, but I am not sure how much does this affect the statistical models. Somehow this should be taken into account in the analyses, not just mentioning it in the discussion. Maybe doing an analysis separated for the northern hemisphere and the rest could be an option to test for this bias in the data.

R/ Accounting for sampling bias in GBIF was also a major concern for us. For this reason, we explicitly accounted for it including the sampling effort variable (based on species inventory completeness, from Meyer et al., 2015) as part of our predictors. As expected, accounting for sampling intensity in GBIF has a strong effect on the observed patterns. Following a suggestion by reviewer 1 we created a bias-controlled map that shows the patterns accounting for the differences in sampling effort across regions. We agree that GBIF is not perfect but is the most complete dataset available for species records and is frequently used in the literature.

Moreover, in the case of invasive species, it is particularly difficult to separate the effect of sampling bias given the uneven distribution of invaders. Alien species are supposed to be more concentrated in areas with stronger human activity and therefore more sampling effort.

2) The use of the IUCN Red List to validate the, that is a list of threatened species that have undergone an extinction risk assessment, but does not contain many species, that are simply not threatened, that contain some of the most invasive species. So the authors claim they

“double-checked the alien status of each occurrence by performing a spatially explicit filtering procedure using available geographic information on species' native ranges...”. I suggest that you explain which proportion of the 717 species were not able to assess with IUCN polygons. I am not suggesting that the work is not correct, but that you should provide a fair assessment of the validity of the results presented.

R/ All 717 species included in the study were assessed with the spatial filter which means that all of them had an available IUCN polygon. It is important to clarify that IUCN polygons are also available for non-threatened species, for example those in the categories of least concern and vulnerable, in which many alien species fall.

3) The concept of mountains is well defined, and mountain ranges have been identified by previous studies throughout the world. But also, it is well established that there are different types of mountain ranges, since some of them are relatively low in altitude and others are higher, referred as “high mountains”. These different types of mountains have very different characteristics that are very relevant for the natural distribution of species (e.g. in the Northern Hemisphere fish are not native from high altitude aquatic ecosystems, but they are native in the lower mountains). I was disappointed by seeing that the species distribution with elevation was left to what is Supplementary Figure 5, that basically is a density plot of richness. It would have been very interesting to see a more in-depth analyses comparing “high mountains” from the rest.

R/ Thanks for the suggestion. Our model includes several predictors that quantify the physical features of the mountains. All of them are intended to describe the mountain types using different properties, like their geometry or topographic complexity. One of the predictors is indeed the elevation range covered by the mountain which seems to mirror to some level the point you stress. On the other hand, although we also considered it interesting to analyze the altitudinal distributions, taking into account the potential impact of geographic uncertainty on elevation assignation in mountainous terrain we were very cautious and avoided conducting further analyses using elevations extracted from the data and limited the exploration to what is now shown in Supplementary Figure 8. Accordingly, we mention in the figure caption that: “*The*

information plotted was extracted from an SRTM elevation layer at 30 arcsec resolution (c. 1x1 km at the equator) using the records here studied. Since these records were obtained from GBIF we expect uncertainties in the elevations, mainly in rough topographies where elevation changes can be strong even in short distances". This figure, by the way, shows a density plot of records not species richness.

Minor comments:

Title: I think the title should be a bit less ambitious. I encourage the authors to include richness, so that people understand that it is the main goal of the paper. You assess the occurrence of the species (with many considerations, as described above) but by no means their spatial distribution. For example, you might have a very small AVR value, but if these species are invasive and have a wide distribution the mountain range in question will have a very bad conservation status.

R/ Thanks for the advice. Following your suggestion, we modified the title by removing the word status, which seems could be wrongly assumed as related to conservation status. As we consider that by using occurrence data we are able to describe as much as possible the spatial distribution of these species and from there infer the richness patterns we also stress this in the new title: *The global distribution patterns of alien vertebrate richness in mountains*

Statistical analyses. Overall, the main GLMM analyses are clear, they provided a repository of the script. However, they have performed a lot of analyses which are not so clearly described. For example, the randomization of alien species composition (L134-141) how was it performed? What about the maps of Figure 1? Can you please double check that all statistical analyses used are clearly cited?

R/ Thanks for the observation. We have added new text to the methods section providing more details on particular analyses when missing, as well as the codes generated to perform them. Nevertheless, some of the info requested was already there, but perhaps overlooked. For example, regarding the randomizations we clearly explained in the original version the following:

“To test whether the flows observed among native and invaded realms are higher or lower than expected by chance, we created a series of null models. We first compiled information on the native realms of roughly 38,000 vertebrate species, to create a global pool of vertebrates, based on the range maps from IUCN and eBird used in the previous sections. Then, we randomized the alien species composition of each realm by resampling from the full pool of global vertebrates the number of species (with their respective native realm) documented as having alien records in each realm. We repeated this procedure 999 times to generate a random distribution of simulated alien compositions on each realm, which we then compared with the observed one to assess the statistical significance of differences. The observed number was considered smaller or greater than expected when it was in or beyond the lower 2.5% or upper 2.5% of the distributions of the 999 random draws, respectively.” Lines 150-160. In addition, in this new version, we added the respective code.

Similarly for the maps in the former Figure 1 (now Figure 2) we explain in lines 132-138 that *“To visualize the global distribution of alien records in mountains, we plotted in the geographic centroid of each mountain the value of species richness accounting for the mountain area (extracted directly from ³⁸). For this, we estimated the values of alien record density by dividing the number of total alien records and those of each taxonomic group reported per mountain with the respective mountain area. Then we mapped this metric for all vertebrates and taxonomic groups separately, as shown in Figure 2.”* Now the code for the creation of these maps is available in the repository.

I wonder why did you use a linear method (Generalized Linear Mixed Model (GLMMs) for analyzing your data, that is basically not linear. Why did not use a Generalized additive model (GAM) (Pedersen, Miller, Simpson, & Ross, 2019; Wood, Goude, & Shaw, 2014) approach that is more robust and allows for modelling non-linear variables? Did you check for linearity of the data and transformed them? I understand the case of area, but what about the other variables?

R/ Thanks for the observations. Indeed, we also adjusted a GAMM, and checked that none of the responses are strongly non-linear. For this reason we opted to use GLMMs as they are more easily interpretable, less prone to overfit, and facilitate our goal of understanding the direction

and magnitude of the effects of each of the studied predictors. It is important to highlight that both overall models (GLMM and GAMM) showed similar results. In this new version we are providing the response curves of the GAMM model in the supplementary material showing this (See Supplementary Figure 4). Moreover, the code used for this analysis is also available in the repository.

Table 1. Sampling effort is referred to sampling completeness in the text. Can you use only one term throughout the manuscript?

R/ Thanks for spotting this. We have modified the manuscript to stick to sampling completeness through the text as well as in the figures.

L221. At this point it would be good to mention that the data is available and the link at the end of the paper. However, why not use a repository that provide a DOI? I think it is a better idea. The Github repository is more for sharing software than actual data. On the other hand, it would be interesting to have a table in the supplement with the list of the different species, so that one may easily check what species are in the dataset.

R/ Thanks for the suggestion. We are currently working on a data paper that will include a full dataset of alien occurrences of the studied taxonomic groups in mountains globally. Moreover, we will provide a workflow to develop user-specific queries by taxonomic group, species, and mountains. In addition, we are developing a shiny app that will be launched with the publication and aims to facilitate visualization and exploration of species lists and the mountains' features directly on an interactive map. This project is advancing well and we are confident that the corresponding manuscript will be submitted in the next few months.

Another aspect is the differences in the number of species within each animal group. Some groups such as amphibians have much less species than the others. Is this because there are not many alien amphibians, or a result of a data bias (being underrepresented) or that there are fewer amphibian species and therefore this is expected? Is the number of alien species a result

of the number of species available or there are differences on the invasiveness of the different animal groups?

R/ It is true that among vertebrates Amphibia is the class with the fewest species documented as alien. However, in detecting alien vertebrate species, in addition to the obvious importance of the available global species pool, the observers also play a relevant role. Given the habits of most amphibian species, their few uses, for example, as food compared to other groups like fishes may also explain such differences. On the opposite end, groups like birds and mammals are likely more easily detectable by most people whether by their daytime activity or large size, respectively.

L250. I think that the chord diagram is not really a network analyses.

R/ We agree and replaced the term “network analysis” with “flow diagrams”

L260. Replace Table 2 to Table 3.

R/ Done.

Figure 2 and the rest of figures and Tables. You cite some papers not following the numbers that are used in the text, which makes it more difficult to find the reference in the reference list. I would suggest to include the number after the reference, even if you cite the author and the year.

R/ Thanks for the suggestion, we changed it accordingly

L310-319. It seems unnecessary to cite estimate [\pm SE] every time. Just at the first case should be enough.

R/ Suggestion accepted.

L444. There are no “taxon” specific models, I would say they are “animal group” model. One may think you did run a model for each taxon.

R/ Thanks, we now call them “vertebrate group models”.

Supplementary Figure 5. Figure S5. Why to names? I wonder why you scale the plots up to 6000 meters while there are no species above 4000 m. I think it would be better to cut the plot at the last elevation.

R/ Thanks, we corrected the title of the figure.

References

Pedersen, E. J., Miller, D. L., Simpson, G. L., & Ross, N. (2019). Hierarchical generalized additive models in ecology: an introduction with mgcv. *PeerJ*, 7, e6876.

Seebens, H., & Kaplan, E. (2022). DASC0: A workflow to downscale alien species checklists using occurrence records and to re-allocate species distributions across realms. *NeoBiota*, 74, 75-91.

Wood, S. N., Goude, Y., & Shaw, S. (2014). Generalized Additive Models for Large Data Sets. *Journal of the Royal Statistical Society Series C: Applied Statistics*, 64(1), 139-155.

doi:10.1111/rssc.12068

RESPONSES TO REVIEWERS

Reviewer #1 (Remarks to the Author):

After careful reading the revised manuscript and the response letter, I am overall satisfied with the new version and the explanations. The manuscript has also been significantly improved. However, I think there are still two main aspects needs to be further strengthened.

First, although the authors argue that this work focuses on the mountain areas and comparison of patterns with other lands is beyond the scope of this study. I am sorry that I cannot completely agree with this opinion. It is necessary to establish a reference point to understand the situation of biological invasion in mountain areas. In fact, many alien species have also invaded in other regions far from mountains. Such comparison will be very necessary and important to provide insights into the pattern and influencing factors of alien species in mountain areas. For example, a recent study exploring the biological invasions on indigenous people's lands (IPLs) also compared the alien species richness on IPLs and other lands after accounting for sampling intensities.

R/ We appreciate the suggestion and acknowledge that such a study comparing mountains and other regions would deliver interesting insights. However, it would result in a completely different manuscript (i.e., tackling different questions). As described in the main text of our manuscript, the questions we are addressing are focused on patterns of alien vertebrate richness in mountains. The primary objective is to understand the drivers shaping variation in richness within this specific habitat. This targeted approach allows for a detailed exploration of ecological patterns and processes that are unique to mountain areas. Moreover, to study biogeographical and macroecological questions within a specific study system in depth, as we perform, is a well-established approach in ecology. For instance, highly cited studies published by leading journals in ecology and invasion science have used this approach to study particular regions without comparing them to other types of regions (e.g., Bellard et al., 2017; Moser et al., 2018; Pysek et al., 2010). The study mentioned as an example by the reviewer has a clear focus on comparing alien richness across different lands, ours is focused on the comparison among mountains across the world. However, we recognize the value in such comparative studies and agree that this could be an interesting direction for future research, potentially building on the foundation laid by the present work.

Second, regarding the role of invasion meltdown in explaining the invasion outcomes, the authors argued that it may have data resolution and analysis circularity issues. Although the country level data is indeed less accurate, but it is the currently only available data and it is

worth to do at least an additional analysis provided in supplementary materials. Regarding the variable circularity issue, I suggest that the authors can use the prior presence of other alien species, which would not require data from the vertebrate response variable.

R/ In the seminal work proposing the invasion meltdown hypothesis, Simberloff & Von Holle (1999) suggested the term ‘invasional meltdown’ defining it as “the process by which a group of nonindigenous species facilitate one another’s invasion in various ways, increasing the likelihood of survival and/or of ecological impact, and possibly the magnitude of impact. Thus, the invasion meltdown hypothesis requires interaction of species, which occurs on the scale of habitats. At the scale and resolution of our study (global extent and covering over 4000 mountains), it is not realistic to gather the required data on species co-occurrence to robustly test this hypothesis. The suggestion to use the prior presence of other alien species as an explanatory variable to avoid data circularity is indeed a thoughtful approach. However, this method also has limitations, primarily because it still requires high-resolution spatial and temporal data on species distributions that is not available at the scale of our study. At the very best we might find some correlation between cross-mountain invasion patterns of different taxonomic groups, but any conclusion on the drivers of this correlation would remain speculative. Given these constraints and the considerable additional effort required to collect and analyze additional data with uncertain outcomes, we decided it would be more prudent to focus our current analysis on the available data. However, our data might provide a basis for testing the invasion meltdown and other hypotheses in the future.

Reviewer #2 (Remarks to the Author):

I reiterate the introductory comments provided during the first round of reviews. Research on alien species in mountains is largely biased towards plants. Accordingly, by looking at alien vertebrate species in mountains this paper contributes to filling a gap and is an important contribution to our understanding of the threat that invasive alien species represent for mountain ecosystems. As such I very much welcome this effort that I see as important. The authors make smart use of existing data on alien species and map them onto mountain ranges to detect spatial patterns in species richness, including in PAs. Additional analyses generate a number of results on the flow of species and on possible drivers.

R/ Thanks again for this very positive and encouraging feedback.

One main comment on the first version of the manuscript was that the discussion was quite shallow. The authors have worked on it to enrich it a bit and I welcome this effort. Besides some smaller comments listed below I would like to note that although the discussion has become better, it remains somewhat unsatisfactory from my point of view. The additions do in fact read a bit as if they were added somewhat quickly, with little attention to the English

and to the logic of some arguments. Moreover, from my point of view the discussion still lacks useful information on what do with those results, on ways forward. It reports and clarifies in light of existing results in other taxonomic group but offers little if any enlightening thoughts or ideas on how to concretely make use of those results in research and management (besides in the context of monitoring). I will argue again for taking the time to think beyond observations in this discussion and to formulate ideas in sharp English.

R/ While we appreciate the suggestion, we would like to stress the limitations of deriving more applied conclusions or management suggestions from a macroecological analysis. We conceived this study in a macroecological framework to provide the big picture of the current situation of alien vertebrates in mountains worldwide, therefore this represents a basic research work and does not aim to guide management guideline. Nevertheless, we now mention this in the manuscript and have made the best of our effort to delve into some additional aspects in the discussion. For example, we have added some arguments and recommendations on the need to keep the human footprint in mountains as low as possible to minimize the spread and establishment of alien vertebrates there. A closing paragraph touching on this is now in lines 530-545 and reads as:

“Our study provides a comprehensive synthesis that, for the first time, presents a global overview of the current situation of alien vertebrates in mountainous regions. While the scale and resolution of our work precludes us from making specific recommendations for individual mountains or species, the patterns we identified underscore critical need of keeping mountains as pristine as possible to minimize the spread of vertebrate invaders. To achieve this, efforts should focus on reducing the human footprint in these areas, where it is increasing due to factors such as rising tourism, as well as novel anthropogenic pressures, for example the ones linked to the accelerated development of renewable energy infrastructure ¹. Similarly, restrictions on the construction of new roads and the development of hiking trails in mountainous regions should also be considered, as these can serve as potential pathways for alien species ^{2,3}. This is especially relevant for tropical mountains which have been historically underrepresented in the global protected area network ⁴ and for mountains in general, as climate change may relief cold-temperature constraints potentially triggering a boost of additional invasions into mountain areas in the near future. We hope that our global-scale analysis will stimulate intensified research on alien mountain invasions to guide the conservation of these peculiar ecosystems.”

I. 41. Statements about how much land surface mountains cover are useless, unless it is made clear that this is based on a given definition of mountains and why that definition is chosen. The 25% is based on the WCMC definition and a priori there is no good reason to apply that definition to work on mountain biodiversity and not that of e.g. Koerner et al, or

Snethlage et al. The authors use the mountain inventory by Snethlage et al., which in its original version is not based on the WCMC definition.

R/ To avoid confusion we have removed the reference to the percentage of land surface covered by mountains in the new version.

I. 54-56. Since ice cover retreat belongs to climate change (cf answers to the comment), it should be mentioned before land-use change, together with climate change.

R/ Done

I.69. "the cases of" can be removed to ease the reading and the sentence modified accordingly. There is a growing body of literature on fish invasions in mountain lakes to which you could refer in more general terms.

R/ Removed

I. 92. In what does the hierarchical approach provide high accuracy? The rivers do indeed help achieve accuracy, but this is unrelated to the hierarchical approach.

R/ To avoid confusion we removed the word "hierarchical" as we agree this is indeed unrelated to the accuracy.

I. 94. The version number is v2.0 and not v.20.

Comment to line 92-93 of the first manuscript: I don't seem to find the sentence you quote in your answer.

R/ Thanks for spotting this. We made the respective edit and added the missing sentence. Please see lines 102-104.

I. 114-117. This sentence is long and unclear. I recommend revisiting it.

R/ We split it into two shorter sentences and tried to improve the clarity. The text (lines 115-118) now reads as: "Following the precautionary principle, we analyzed DASCO's records, preferring overestimation to underestimation. This approach reduces the risk of incomplete descriptions of alien species spread by also considering casual populations that later may exert impacts on biodiversity and ecosystem services."

I. 129-130. You mean you discarded from the vertebrate records' subset those records falling The edits are not very clear.

R/ Yes. We modified it as suggested, please see lines 129-131: “For each species, we then discarded from the DASCO’s vertebrate records’ subset, those records falling within the native range polygons.”

I.138. You mention Figure 2 before Figure 1 in your manuscript.

Figure 2. The centroids are hardly visible. This Figure needs to be revisited. Are the centroids really needed?

R/ We removed the reference to figure 2 from the methods section. All the main figures are cited for the first time in the results section and are mentioned in the order they appear in the manuscript.

The centroids are strongly needed because otherwise the high number of records spread across some mountains would saturate the figure. Using the centroid, we were able to use a single point per mountain which depending on its darkness shows the number of observations in the respective mountain.

I. 175-178. This points was raised before and I am still at odds with the approach taken. Why did you not intersect your PA layer with the mountain polygons to get only the proportion of PAs that is truly in mountains? I also don’t quite understand the argument you propose for using the full PA area when only a fraction is in mountains. I acknowledge the answer you gave but still don’t understand why you don’t account for the mountainous fraction of PAs. Why would the full size of the PA and not the size of the mountainous fraction have an effect? Would the edge effect not apply equally if you considered the mountainous fraction?

R/ To address this point we have now calculated the surface of each studied PA that is distributed within mountains. After this exercise we reach similar results when correcting for area. Nevertheless, in this new version we opted to update the respective figure (Fig. 4, panel C). In addition, we also changed the text in methods specifying that we considered only the surface area of each PA located in mountains for the correction (line 331), and removed the text justifying the use of the full area. Similarly, we modified the related text in results and discussion.

I.202 (former line 176). the authors argue that rough terrains provide a wider suite of habitats, which I get but does a mean value for an entire mountain range make sense? The roughness could be high in some part of the mountain ranges and very limited elsewhere. For larger mountain ranges, a mean value seems of little relevance.

R/ We understand the point raised by the reviewer. However, we must note that the alien species numbers we analyze refer to the entire mountain area, and the use of an average

roughness value serves to characterize the overall diversity of habitats in the region. While analyzing this at a higher spatial resolution could certainly be possible, it would still require aggregating at the regional level for inclusion in our analyses.

I.257: what do you mean with “considerable”? Is it more or less than expected, than some sort of mean? This is subjective.

R/ We removed the word considerable and used the actual number as reference to make it less subjective and more descriptive.

I. 257-259. What is the value of such a statement if you don't say what mountains those are?

R/ The mountains can be seen in the Supplementary figure 5. Therefore, we moved the reference to this figure to fit the statement questioned.

I. 259. Why use “still”?

R/ We want to highlight that with and without the sampling effort correction, mountains in Europe and US are the ones with the highest number of alien species reported. Anyway, we have removed the word “still” to fully avoid confusion.

I. 259-262. You talk about positive deviations (from what?) and residuals. What are those residuals, how are they calculated, and what do they represent?

R/ Residuals are the differences between an observed and a predicted value in a regression analysis. Positive residuals represent deviations where the observed value is higher than expected. In the context of our results, we refer to this to highlight that despite the correction by sampling completeness, mountain ranges in Europe and the USA are still the ones with the highest alien richness documented.

I. 261-262. I don't quite understand this sentence: you are saying that only a few mountains in Mexico, China, Australia, and Brazil are among those with positive residuals above 10 but below 25. What or where are the "other mountains" (since you say “among those” it means that there are others).

R/ Following your recommendation, we re-wrote this sentence as: “A few mountains in Mexico, China, Australia, and Brazil have positive residuals above 10 but in all cases below 25 (see Supplementary Figure 5).” Lines 260-261.

I. 267. “on the contrary” and not “contrary”

R/ Suggestion added.

I. 284-288. You refer to Olsen’s ecoregions in this section although in your methods you introduce them only in view of reporting on the global flows of species.

R/ Thank you for this comment. Please note that as written in the manuscript (lines 141-142), we used Olson’s biogeographic realms delimitation, not ecoregions. Here in specific we also referred to the realms and provide examples of more specific regions particularly within the Neotropics aiming to provide a more accurate description.

Figure 5: why use a different colour for the taxonomic groups then in figure 4?

R/ In the new version we have modified the colors to make it match with those used in other figures for the taxonomic groups.

I. 462-464. please revise the English

R/ We modified the text for more clarity. Now it reads: “Interestingly, although only 7% of all records refer to the other three taxonomic groups, these 7% represent more than one-quarter of the species reported in protected areas (PAs) in mountain regions.” Lines 461-463

I. 468-471. The use of “on the other hand” and of “also” is unjustified. Also “reduced” as compared to what?

R/ We removed both.

I. 473-475. Would the human presence necessarily increase the detection probability? This argument seems somewhat lost there. Do you have any reference or context that you could add to enrich this statement?

R/ In the previous sentence, indeed we provide a reference supporting the fact that: “Previous studies have shown that, at least in Europe, human accessibility is a major predictor of alien richness in PAs ⁷⁸”. In the lines here criticized we suggest that our results seem to adhere to this trend because when “Correcting for the area, our data shows that this trend is even more evident as small and accessible protected areas show the highest numbers of both records and species richness relative to their area”. Nevertheless, we remain cautious about this potential conclusion and add that “It is still an open question what the relative contribution of human presence is in increasing both the detection probability and the introduction rates of alien species”. In light of this, we consider that

our argumentation is correct and clear and would like to keep it as it is now, only including a new sentence where we provide a couple of references supporting the fact that population density and accessibility are good predictors of detection probability. Lines 472-474 now read as: “While accessibility and human population density have proven to be good predictors of detection probability in different taxonomic groups⁷⁹⁻⁸¹, ...”

I. 484-488. Please reformulate by cutting this sentence in two. It is too long.

R/ Done

I. 488-490. Please revise the English.

R/ We modified for more clarity as: “While broad elevational ranges can create diverse climatic conditions, offering suitable environments that facilitate the establishment of multiple species, they can also present significant physical and physiological barriers, limiting the dispersal and survival of others⁸⁴” **Lines 487-490.**

I. 491. What do you mean with “vertebrate group models”?

R/ Models fitted separately for each group of vertebrates (i.e. fishes, amphibians, reptiles, birds and mammals). To improve clarity, we have modified the text as:

“By deconstructing the general picture into models separately fitted for each vertebrate group (i.e. fishes, amphibians, reptiles, birds and mammals), we found that the effects of the tested drivers vary across different taxonomic groups.” **Lines 491-493**

I. 493-494. This should be a sentence on its own.

R/ Done.

I. 491-494. Do you have some references from other studies to make your point with an example?

R/ We have now provided a proper reference on the role of life history traits in animal invasions.

I. 494. Please use something else then “concerningly” to start the sentence. And why do you need to use “still”?

R/ We use now “nevertheless” and removed “still” (Line 496).

I. 497-499. Please revise the English and specify what repercussions you are talking about

R/ Now we clarify that this is related to ecological repercussions. In the new version, lines 499-500 read as: “This may lead to the replacement of native species by alien ones likely resulting in strong ecological repercussions in vulnerable regions like mountain ranges.” In the next sentence we highlight local extinctions and biotic homogenization as some of these repercussions.

I. 501. What specifically do you mean with “consequences for conservation”? Such a statement is of little interest if you don't expand on it (cf my general comment on the discussion)

R/ Extinctions itself as well as biotic homogenization are major consequences of the turnover generated by the loss of native species and the increase of introduced ones. Since the reviewer considers the statement “consequences for conservation” of “little interest” we decided to remove it and shortened the sentence.

I. 510. The “play an important role in” is unnecessary

R/ Removed.

I. 512-515. Making the point that the probability of detection increases with sampling effort in undersampled regions is not offering a particularly interesting insight. Moreover, the link between this argument and the fact that diverse environmental conditions could offer more opportunities is unclear. Please revisit this argument.

R/ We have removed this entire sentence as we agree that it does not contribute much to the discussed point.

Reviewer #3 (Remarks to the Author):

I have reviewed the changes made in the paper and the responses given by the authors, and I am pleased to see they have satisfactorily responded to all my corrections.

R/ We are very happy about the positive assessment of our revised manuscript. Thanks.

References cited:

Bellard, C., Rysman, J. F., Leroy, B., Claud, C., & Mace, G. M. (2017). A global picture of biological invasion threat on islands. *Nature Ecology and Evolution*, 1(12), 1862–1869. <https://doi.org/10.1038/s41559-017-0365-6>

Moser, D., Lenzner, B., Weigelt, P., Dawson, W., Kreft, H., Pergl, J., Pyšek, P., van Kleunen, M., Winter, M., Capinha, C., Cassey, P., Dullinger, S., Economo, E. P., García-Díaz, P., Guénard, B., Hofhansl, F., Mang, T., Seebens, H., & Essl, F. (2018). Remoteness promotes biological invasions on islands worldwide. *Proceedings of the National Academy of Sciences of the United States of America*, *115*(37), 9270–9275.

<https://doi.org/10.1073/pnas.1804179115>

Pyšek, P., Jarosik, V., Hulme, P. E., Kuehn, I., Wild, J., Arianoutsou, M., Bacher, S., Chiron, F., Didziulis, V., Essl, F., Genovesi, P., Gherardi, F., Hejda, M., Kark, S., Lambdon, P. W., Desprez-Loustau, M.-L., Nentwig, W., Pergl, J., Pobljsaj, K., ... Winter, M. (2010). Disentangling the role of environmental and human pressures on biological invasions across Europe.

Proceedings of the National Academy of Sciences of the United States of America, *107*(27), 12157–12162. <https://doi.org/10.1073/pnas.1002314107>

Simberloff, D., & Von Holle, B. (1999). Positive interactions of nonindigenous species: invasional meltdown? *Biological Invasions*, *1*, 21–32.